# Adapting Language Models to Produce
# Good Class Probabilities for Classification Tasks

**Lautaro Estienne**  *lestienne@fi.uba.ar*
*School of Engineering, UBA, Argentina*
*ICC, CONICET-Universidad de Buenos Aires, Argentina*
*LISN, CNRS, Université Paris-Saclay, France*
*International Laboratory on Learning Systems, Canada*

**Matías Vera**  *mvera@fi.uba.ar*
*School of Engineering, UBA, Argentina*
*CSC, CONICET, Argentina*

**Elizabeth Fons**  *elizabeth.fons@jpmorgan.com*
*J.P. Morgan AI Research, UK.*

**Elena Kochkina**  *elena.kochkina@jpmorgan.com*
*J.P. Morgan AI Research, UK.*

**Pablo Piantanida**  *pablo.piantanida@cnrs.fr*
*International Laboratory on Learning Systems, Canada*
*Mila - Quebec AI institute, Canada*
*CNRS Université Paris-Saclay, France*

**Luciana Ferrer**  *lferrer@dc.uba.ar*
*ICC, CONICET-Universidad de Buenos Aires, Argentina*

**Reviewed on OpenReview:** *https://openreview.net/forum?id=VVneIp69GR*

## Abstract

Large generative language models (GLM) provide a versatile tool for solving a wide variety of natural processing tasks. GLM responses, though, are provided in the form of text, without an indication of the model's confidence in the answer. This limits the usability of these models on high-risk applications where decisions made based on an incorrect answer can have severe consequences. In this work, we focus on the problem of generating class posterior distributions for text classification tasks like sentiment, news category and intent classification. These posteriors can be used for decision making and as interpretable scores for the user. We show that the naive approach for computing posteriors based on the token posteriors produced by the GLM results in extremely poor posteriors. We then explore different adaptation approaches for improving the quality of posteriors, focusing on low resource scenarios where a small amount of data is available for adaptation. We show that parameter-efficient supervised fine-tuning (SFT), while providing large gains in terms of decision quality, produces suboptimal posteriors due to overfitting. To address this problem, we propose an approach that combines SFT and post-hoc calibration (PHC) using a three-stage training strategy, improving the quality of both posteriors and categorical decisions.

## 1 Introduction

Modern neural language models provide an effective, adaptable and scalable tool for solving many natural language processing tasks. In particular, large generative language models (GLMs) like LLaMA 3 (Grattafiori

et al., 2024), Qwen 2.5 (Yang et al., 2024) or Phi-4 (Abdin et al., 2024), are currently being used for a variety of complex natural language understanding tasks, showing outstanding performance on benchmarks related to reading comprehension, summarization, information retrieval, and generative question-answering (Narayan et al., 2018; Zellers et al., 2019; Khattab et al., 2023; Omar et al., 2023; OpenAI et al., 2024; Wei et al., 2021). While surprisingly good out-of-the-box performance can often be achieved with these models, they are generally unable to produce reliable measures of uncertainty for their answers, as shown in a large number of works (Lin et al., 2024; Cole et al., 2023; Estienne et al., 2023; Kuhn et al., 2022; Zhao et al., 2021). This limits the usability of these models for high-risk tasks where only very high confidence answers should be accepted (Zhang et al., 2021; Yoon et al., 2020).

In classification tasks, which are the focus of this work, optimal decisions are made based on class posterior probabilities using Bayes decision theory (Hastie et al., 2001; Bishop, 2007; Ferrer, 2024). For example, for the common case where all errors are assumed to cost the same, Bayes decisions correspond to the class with the highest posterior. The quality of the decisions then directly depends on the quality of the posteriors. Further, the posterior for the selected class is a measure of the confidence with which the system made the decision. This confidence score is valuable information that can be provided to the end user along with the decision. The goal of this work is to develop an approach for adapting GLMs for a downstream classification task to produce good-quality posterior probabilities for decision making and for interpretation by the end user. We use the phrase "good-quality posteriors", to refer to posteriors that can be trusted for decision making with Bayes decision theory (see, for example, Winkler & Murphy (1968)). A system's posteriors are better quality than another's if they result in better Bayes decisions.

Various approaches have been proposed for adapting GLMs to a specific domain. Standard techniques such as adding examples in the prompt to illustrate how the task should be solved or asking for a detailed explanation before producing an output are part of the family of methods called *prompt engineering* (or *prompting*, for short) (Liu et al., 2023). These methods have relatively low computational cost during the development phase because they do not require updating the parameters of the model. On the other hand, during deployment, they have the disadvantage of consuming tokens from the available prompt buffer and slowing down inference. Furthermore, the performance of the resulting system has been shown to strongly depend on the specifics of the prompt, making it a somewhat fragile approach (Zhao et al., 2021; Estienne et al., 2023). Another alternative for adapting a GLM to a downstream task or domain is supervised fine-tuning (SFT), which requires annotated data for the domain of interest. Due to the large size of modern GLMs, updating all of their parameters is computationally expensive and requires large amounts of adaptation data. To address this issue, a variety of parameter-efficient fine-tuning (PEFT) methods were proposed in the literature, among which LoRA (Edward J Hu et al., 2022) is currently the de-facto standard.

Although SFT methods are very effective for improving the performance of the model on the task of interest, our experiments show that they do not ensure that the scores assigned to the predictions are good posterior probabilities for the task. Motivated by these results, we explore the use of post-hoc calibration (PHC) methods, which are approaches specifically designed to transform scores produced by a classification system into reliable posterior probabilities. In particular, we consider the family of affine logistic regression methods, which is widely used for this purpose and very effective.

We evaluate the systems under comparison using two different metrics. To assess the quality of categorical decisions, we use the standard error rate, which assumes all errors to be equally costly. To assess the quality of posteriors we use proper scoring rules, as recommended by Ferrer & Ramos (2025). In contrast, recent machine learning papers concerned with this problem frequently use the expected calibration error (ECE) as a quality measure (Guo et al., 2017; Si et al., 2022; Ao et al., 2023). Yet, as discussed by Ferrer & Ramos (2025), the ECE and other calibration metrics do not adequately address the problem of assessing the value of posteriors since calibration only reflects one aspect of their performance (Gneiting & Raftery, 2007; Kull & Flach, 2015; Ferrer & Ramos, 2025). Hence, we instead use the cross-entropy, which, as the expectation of a proper scoring rule, is specifically designed for the problem of measuring the quality of posteriors (Winkler & Murphy, 1968; Gneiting & Raftery, 2007; Ferrer & Ramos, 2025).

## 1.1 Summary of our contributions

In this work we propose an effective way to combine SFT and PHC for text classification systems based on GLMs. The proposed approach, described in Section 5.3, provides the best quality posteriors (as measured by cross-entropy) and, as a consequence, also the best quality decisions (as measured by error rate) compared to using only SFT or PHC. Importantly, our results show that a careful training strategy is required to combine the two methods in order to achieve simultaneously low cross-entropy and error rate. Naively training PHC on the same adaptation data used to train SFT gives the same performance as SFT alone, while splitting the adaptation data for SFT and PHC training is suboptimal. In addition, we show that temperature scaling–the most common calibration approach in current machine learning literature–is often ineffective for calibration of GLM scores. A simple generalization to an affine transformation provides significantly better results, sometimes competitive with those obtained with SFT alone. These results are shown in sections 6.2, 6.3, 6.4 and 6.5. Finally, in Section 6.6, we show that, after adaptation with the combined approach, the results from a small LLaMA model are similar to those from a significantly larger Qwen model. These results suggest that suboptimal models become competitive after adaptation with our proposed method, though this conclusion will have to be verified by experimenting with a larger number of GLMs. The code needed to replicate the results in this work, along with output scores for all methods under comparison, can be found at `https://github.com/LautaroEst/llmcal`.

## 2 Previous Works

Many methods for aligning a GLM to a specific task of interest rely purely on Input-Output prompting techniques (Brown et al., 2020). These methods adapt a generic GLM to a given task by carefully designing the prompt instructions (Raffel et al., 2020), selecting examples that illustrate how to solve the task (Brown et al., 2020) or retrieving additional information to provide necessary context (Khattab et al., 2023). More elaborate methods, such as Chain-of-Thought (CoT) (Wei et al., 2022), Tree of Thoughts (Yao et al., 2023) or BetterTogether (Soylu et al., 2024), optimize the prompt by iteratively generating and refining context to improve task performance. Notably, prompting techniques are limited by the finite context length and increase runtime during deployment with respect to simple prompts. In this work, we focus on simple prompts for clarity and efficiency, while noting that our proposed method is compatible with—and can be layered on top of—more elaborate prompting strategies such as the ones mentioned above.

SFT methods can adapt a single model to one or multiple tasks using in-domain adaptation data (Wei et al., 2021). Given the large size of modern GLMs, fine-tuning all of their parameters requires a substantial amount of data and computational resources (Chung et al., 2024). To mitigate this problem, parameter-efficient fine-tuning (PEFT, Lialin et al., 2024) techniques have been proposed in the last few years, including additive (Lester et al., 2021; Houlsby et al., 2019), selective (Donahue et al., 2014; Gheini et al., 2021), and reparametrization (Aghajanyan et al., 2021; Liu et al., 2024) variants. In particular, LoRA (Edward J Hu et al., 2022), a reparametrization method, has shown exceptional performance across numerous tasks. LoRA reduces the number of trainable parameters by replacing the linear operations inside the network with a low-rank approximation based on the hypothesis that the difference between the new and the old parameters lies in a low-rank space. Importantly, the adaptation cost of SFT methods is restricted to the fine-tuning stage. The resulting model can run faster than those that use prompt-engineering because the prompts do not need to describe the task. On the other hand, even when using PEFT methods, adapting a GLM requires more data than what is needed for prompting (Mosbach et al., 2023; Zhang et al., 2023; Edward J Hu et al., 2022). We will use a LoRA-based SFT method which provides strong empirical performance and a favorable balance between training cost and adaptation effectiveness.

For the specific case of text classification tasks, another family of adaptation approaches consists of introducing a PHC transformation to convert the posteriors generated by the model into new posteriors that are better matched to the task of interest. For example, Zhao et al. (2021) proposed transforming the posterior for each class by dividing it by the prior for that class, estimated by running the GLM with a content-free query. Similarly, Fei et al. (2023) propose dividing the posterior for each class by the average posterior for a query consisting of in-domain random words. These works implicitly assume that the prior class distribution for the task of interest is uniform. Since this assumption holds approximately or exactly for many text

classification datasets, the methods work quite well for those cases. In our prior work (Estienne et al., 2023), we formalized the problem of bias mitigation and showed that superior results can be obtained compared to the approach proposed by Zhao et al. (2021) by shifting the log posteriors with a bias term trained by minimizing the cross-entropy on adaptation data. Further, we showed that the bias term can also be successfully estimated without annotated data, relying only on knowledge of the class priors for the task of interest. In a recent work on PHC for NLP, Spiess et al. (2024) use a Platt scaling calibration trained on labeled data to determine whether a code generated by a GLM is correct or not. In this work, we explore various PHC approaches and combine them with SFT, leveraging their ability to produce better posteriors while incurring negligible additional computational cost during inference.

A few works on text classification with LLMs focus on the problem of producing good quality posteriors, in addition to the categorical decisions. For instance, Li et al. (2025b) propose to mitigate the bias of the model due to each part of the prompt – hypothesis and premise – by normalizing the sequence posterior with the one obtained by prompting the model with each of those parts separately. Alternatively, Jiang et al. (2023) produce class posteriors by averaging the sequence posteriors over an ensemble of prompts created as minor modifications of the original prompt. In other cases, the posteriors or confidences are generated by explicitly asking the LLM to verbalize these probabilities (Tian et al., 2023; Xiong et al., 2024) or by sampling the model multiple times and then counting the frequency of each class (Gal & Ghahramani, 2016; Wei & Zou, 2019). All those methods are unsupervised, not requiring any data for the task. When some in-domain data is available, recent evidence indicates that supervised methods like SFT or PHC can result in better performance. For instance, in our previous work (Estienne et al., 2023), we showed that PHC models trained with a small amount of in-domain data outperformed unsupervised methods. Additionally, Li et al. (2025a) show that fine-tuning-based approaches perform systematically better than in-context learning-based approaches. Finally, a PHC method proposed by Shen et al. (2024) was also shown to produce higher-quality outputs in comparison with unsupervised methods. For this reason, in this work we focus our efforts on supervised approaches.

The metrics used to assess quality of the text classification systems are usually focused on the quality of the categorical decisions and include accuracy, error rate, F1-score or precision/recall (Yin et al., 2019; Schopf et al., 2023; Gera et al., 2022; Gretz et al., 2023). Works that are interested in the quality of the posteriors usually report performance in terms of calibration error (Desai & Durrett, 2020; Kapoor et al., 2024; Stengel-Eskin et al., 2024) which, as mentioned above, is a problematic practice since these metrics do not adequately reflect the quality of the posteriors. Very few works on text classification report performance in terms of PSRs and it is usually done in addition to ECE (Xie et al., 2024; Tian et al., 2023), often leading to ambiguous conclusions. For example, in Tables 2, 3, and 5 of the work done by Tian et al. (2023) we can see that the Brier scores (which is the expectation of a PSR) for the TriviaQA dataset are best for the baseline posteriors, but the ECE is best for the proposed calibration methods. This happens because the calibration methods degrade discrimination with the unintended effect that the overall quality of the posteriors is degraded. Faced with the need to select one of the two systems, one should choose the one with the lowest Brier score, not the one with the lowest ECE since those system's posteriors are poorer. Therefore, in this work we assess the quality of the posteriors using proper scoring rules (PSRs). For completeness, we include results for ECE and another calibration metric called relative calibration loss (Ferrer & Ramos, 2025) in Appendix C.1.

## 3 Computing text classification posteriors using GLMs

There are many ways to use GLMs for downstream tasks. The most common way is to simply prompt them with a question or instruction and have them generate an answer. In some scenarios, though, this approach may produce answers that do not satisfy the requirements of the task. One such case is text classification, where the goal is to annotate a certain text with a class label selected among a pre-defined set of possible classes. A GLM used as a free-text generator may not necessarily respond with one of the valid classes for the task. Further, with this approach, the GLM does not produce posterior probabilities for the classes which, as we have argued, may be a requirement for high-risk tasks. In this section, we describe a general and principled way to use GLMs for producing class posteriors for text classification tasks.

```
Prompt ρ(x, y₂)
<|im_start|>system
Determine the category of the article given by the user.<|im_end|>
<|im_start|>user
The Blair-Caldwell African American Research Library is a
branch of the Denver Public Library in Denver, Colorado,
in the United States that serves the Five Points
neighborhood. [...] <|im_end|>
<|im_start|>assistant
Educational Institution
```

- ■ Prompt template
- ■ Sample **x**
- ■ Class identifier

Figure 1: Example of a sequence $\rho(\mathbf{x}, y_2)$ for the Qwen2.5 model for a DBPedia sample, where $\mathbf{x}$ is the sample text taken from the dataset, and $y_2$ is the label corresponding to class 2 of the DBPedia dataset. The token sequence used to compute the posteriors, $a_2$, is the part that identifies the class and is highlighted in dark purple ("Educational Institution," in this example).

The first step is to determine an appropriate token sequence with which to query the GLM. In this work, we use the procedure formalized by Liu et al. (2023), where a model- and task-dependent template $\rho(\cdot)$ is used to create the sequence of tokens that make up the full sentence (a prompt followed by the answer). That is, given an input text $\mathbf{x}$ to be classified and a class $y_k \in \mathcal{Y} \triangleq \{y_1, \ldots, y_K\}$, we create a token sequence, $(p, a_k) \triangleq \rho(\mathbf{x}, y_k)$ which is a concatenation of the prompt $p$, consisting of the input text $\mathbf{x}$ and some additional instructions, and the answer $a_k = (a_k(1), \ldots, a_k(M_k))$, representing a name for the class $y_k$ (see Figure 1 for an example). Details on the token sequence used for each dataset and each model can be found in Appendix A.

Given a sample text $\mathbf{x}$ we can then create a prompt $\rho(\mathbf{x}, y_k)$ for each possible class, $y_k$, and run the GLM to obtain the posteriors for each token $a_k(m)$ in the answer conditioned on the previous tokens, $P_{\mathrm{LM}}(a_k(m) \mid a_k(m-1), \ldots, a_k(1), p)$. Then, we can compute the posterior for the answer given the prompt as

$$P_{\mathrm{LM}}(a_k \mid p) = \prod_{m=1}^{M_k} P_{\mathrm{LM}}(a_k(m) \mid a_k(m-1), \ldots, a_k(0), p). \tag{1}$$

where $a_k(0)$ is an empty sequence. Finally, we can define a *task* probability $P_{\mathrm{Task}}(y_k \mid \mathbf{x})$ for the class label $y_k \in \mathcal{Y}$ given the input $\mathbf{x}$ as a normalized version of $P_{\mathrm{LM}}(a_k \mid p)$ that sums to one over all the valid classes:

$$P_{\mathrm{Task}}(y_k \mid \mathbf{x}) \triangleq \frac{P_{\mathrm{LM}}(a_k \mid p)}{\sum_{j=1}^{K} P_{\mathrm{LM}}(a_j \mid p)} \tag{2}$$

This is a natural way of using the score produced by the model to obtain a probability for each class. In particular, if each class is represented by a single token, Equation (2) is equivalent to applying the softmax operator to the vector obtained by stacking the logits of the tokens corresponding to each class. However, Equation (2) contemplates the general case where the classes may be represented by a sequence of more than one token.

This simple approach allows us to work around the problem posed by free-text generation, whereby two different sequences of tokens may correspond to the same class. For example, for the polarity classification task, when prompted with the instruction 'Determine if the following review is positive or negative', the GLM may respond 'The review is negative' or just 'Negative'. These two answers would result in different posteriors. We could attempt to force the GLM to produce only the class name in the answer by adding instructions in the prompt. Yet, this would result in long prompts, specially when there are a large number of classes or when each class has a long description, and it would not fully guarantee that the GLM will comply. Instead, we simply query the GLM with the $a_k$ sequence, with the desired answer included and use the token posteriors of the answer to compute Equation (2).

## 4    Proper scoring rules for assessment of classification systems

In this work, we follow the literature on strict proper scoring rules (SPSRs), which were proposed decades ago for the assessment of the quality of posterior distributions (Winkler & Murphy, 1968; Gneiting & Raftery, 2007; Bröcker, 2009). The proper scoring rule (PSR) concept is founded on the assumption that rational decisions (categorical or otherwise) should be made by minimizing the expectation of a cost function of interest. The cost function assigns a numeric value to every possible decision given the true class of the sample. Given a posterior distribution over the classes for a given sample, the optimal decision is the one that minimizes the expectation of the cost function with respect to that distribution (Duda et al., 2001). The cost of this optimal decision, called Bayes decision, defines a PSR (Dawid & Musio, 2014; Brummer, 2010). In other words, a PSR is the cost of the optimal decision for the given posterior distribution.

PSRs satisfy an essential property: their expectation with respect to a given reference distribution is minimized when the posterior under evaluation coincides with this reference. When the minimum is only attained for the reference distribution, the scoring rule is called strictly proper. Strictly proper scoring rules (SPSRs) encourage the distribution under evaluation to be close to the reference distribution across the full simplex. For further details and formal and conceptual explanations on PSRs, we refer the reader to the work by Dawid & Musio (2014), Brummer (2010), and Ferrer & Ramos (2025), among many others.

Non-strict PSRs can be constructed by selecting a cost function for categorical decisions, making Bayes decisions for that cost function, and then measuring the cost. The most widely used non-strict PSR is the error rate of decisions made with the argmax rule, i.e., by choosing the class with the largest posterior.

Interestingly, strict PSRs can be constructed by integrating over a family of non-strict PSRs with non-zero weights across the full probability simplex (Gneiting & Raftery, 2007; Brummer, 2010), allowing for an intuitive understanding of the strictness property (Ferrer & Ramos, 2025). Different SPSRs can be obtained by changing the relative weight of each individual cost function within that integral. One example of an SPSR is the Brier loss, for which the expectation over the data is called Brier score (BS) (Brier, 1950), a commonly used metric in some medical applications (Huang et al., 2020; Van Hoorde et al., 2015). Another SPSR is the negative log-loss (NLL). The cross-entropy (CE) is the expectation of the NLL, a metric widely used as an objective function for training deep neural network models, including GLMs (Wei et al., 2021; Raffel et al., 2020; Chung et al., 2024). In this work, we will mostly use CE instead of BS because the former penalizes extremely wrong posteriors more heavily. While Brier loss has a maximum penalty of 1, the NLL can be infinite when a posterior of 0 is assigned to the correct class. We believe this is a desirable characteristic for high-stakes applications where errors may have extreme consequences. For completeness, results for BS are also included in Appendix C.1.

While the CE is perhaps the most widely used loss for model training, it is much more rarely used as an evaluation metric. This may be partly due to the fact that its values are hard to interpret for being unbounded. Yet, the CE can be normalized to make it easily interpretable. Assuming an evaluation dataset with $N$ samples, $\{(\mathbf{x}^{(i)}, y^{(i)})\}_{i=1}^{N}$, and posteriors $\mathbf{q}^{(i)} = (q_1^{(i)}, \ldots, q_K^{(i)})$ for each of those samples, with $q_k^{(i)} \triangleq P_{\text{Task}}(y_k \mid \mathbf{x}^{(i)}) \,\forall k = 1, \ldots, K$, the normalized CE (NCE for short) is given by:

$$\text{NCE} = \frac{\sum_{i=1}^{N} \sum_{k=1}^{K} \mathbb{1}_{\{y^{(i)} = y_k\}} \log q_k^{(i)}}{\sum_{k=1}^{K} N_k \log(N_k/N)}, \tag{3}$$

where $N_k$ is the number of times the class $y_k$ appeared in the evaluation set (Ferrer, 2024). The normalization is done by dividing by the CE of the best naive system, one that always outputs the prior distribution of the classes, having no access to the input sample. This CE turns out to be the entropy of the prior distribution. Systems with NCE values close to 1.0 have a performance comparable to that of the best naive system.

In this work, as is usually done in machine learning literature, categorical decisions are made by selecting the class with the highest posterior probability. These decisions, commonly referred to as argmax decisions, can be shown to optimize the error rate, ER (or, equivalently, the accuracy which is given by one minus the error rate). The ER assumes that all incorrect decisions have the same cost. As mentioned above, when decisions are made with the argmax rule, the resulting ER is also an expected PSR, although not a strict one. As with the CE, the ER can be normalized by the ER of a system that makes always the same decision

based solely on the class priors, without access to the input sample. The normalized error rate (NER) of argmax decisions is given by:

$$\text{NER} = \frac{\sum_{i=1}^{N} \mathbb{1}_{\{\text{argmax}_k q_k^{(i)} \neq y^{(i)}\}}}{\sum_{k=1}^{N} \mathbb{1}_{\{\text{argmax}_k N_k \neq y^{(i)}\}}}, \tag{4}$$

Importantly, although both metrics are PSRs, a low NER does not necessarily imply a low NCE. As we will show in the experimental section, while the system's outputs may be good for making argmax decisions, resulting in a low NER, they may still be poor posterior probabilities of the classes given the input, resulting in a high NCE. Conversely, a lower NCE does not directly imply a lower NER, though, in practice, this is very often the case. For these reasons, it is important to report both metrics when the quality of the posteriors is of interest, in addition to the quality of the decisions.

## 5 Adapting GLMs to produce good-quality posteriors

In this section we describe the methods we explored to adapt a GLM to a text classification task. We divide the methods into three categories: (1) supervised fine-tuning (SFT), (2) post-hoc calibration (PHC) and (3) combinations of the two approaches.

### 5.1 Supervised fine-tuning (SFT)

In this work, the LoRA method is implemented for fine-tuning (Edward J Hu et al., 2022) using a dataset of sequences created as described in Section 3 from the adaptation samples for the task of interest. We believe the main conclusions from this work should hold for alternative variants of LoRA and other SFT approaches. We leave this exploration for future work.

The optimization loss for SFT is given by the token-level CE corresponding only to the tokens of the answer. That is, given an adaptation set $\{(p^{(i)}, a^{(i)})\}_{i=1}^N$ where each sample $i$ consist of a sequence of tokens $p^{(i)}$ (the input prompt) and a sequence of tokens $a^{(i)} \in \{a_1, \ldots, a_K\}$ (the desired answer), the training loss is computed as

$$\mathcal{L}_{\text{LM}}(\boldsymbol{\theta}) = -\frac{1}{\sum_{i=1}^{N} M_i} \sum_{i=1}^{N} \sum_{m=1}^{M_i} \log P_{\text{LM}}(a^{(i)}(m) \mid a^{(i)}(m-1), \ldots, a^{(i)}(0), p^{(i)}; \boldsymbol{\theta}) \tag{5}$$

where $M_i$ is the number of tokens in answer $a^{(i)}$, $\boldsymbol{\theta}$ is the set of trainable parameters, and $a^{(i)}(0)$ is an empty sequence. We also considered computing the loss over the tokens for the full sentence, the prompt and the answer. We found both versions to give similar results, with the one computed only over the answer being faster to compute. For that reason, all experiments in this work are done with the loss computed only over the answer.

When training or fine-tuning large models, it is often effective to use regularization. This prevents overfitting, which is particularly important when the amount of training data is small, as in our scenario of interest. The LoRA approach to fine-tuning can be seen as a regularization approach since it greatly reduces the amount of trainable parameters with respect to full fine-tuning (Biderman et al., 2024). Yet, given that our results showed that models fine-tuned with LoRA still suffer from overfitting, we compared the following approaches:

- **SFT**: Model fine-tuned with LoRA and no further regularization, trained until convergence of the training loss using the full adaptation set.

- **SFT-ES**: Model trained with LoRA by leaving 30% of the adaptation set for validation, and applying early stopping (ES) based on the CE loss on this set.

- **SFT-ES-retrained**: Model trained on the full adaptation set for a number of steps equal to the optimal number obtained for the SFT-ES model. This approach requires two fine-tuning steps, one to obtain the optimal number of steps using 70% of the data for adaptation and one to obtain the final model using 100% of the adaptation data.

- **SFT-L2**: Model fine-tuned as the SFT one above but with a training loss given by $(1-\lambda)\mathcal{L}_{\text{LM}}(\boldsymbol{\theta})+\lambda||\boldsymbol{\theta}||_2$ where $\lambda \in [0,1]$ and $||\cdot||_2$ is a regularization term given by the L2 norm of the model's parameters.

- **SFT-LS**: Same as SFT-L2 but using a label smoothing regularization term as in (Meister et al., 2020) instead of an L2 term. The loss is then given by $(1-\lambda)\mathcal{L}_{\text{LM}}(\boldsymbol{\theta}) + \lambda\mathcal{L}_{\text{LS}}(\boldsymbol{\theta})$ where $\lambda$ is again a fixed scalar between 0 and 1 and $\mathcal{L}_{\text{LS}}(\boldsymbol{\theta})$ corresponds to the Kullback-Leibler distance between the posteriors distribution over tokens produced by the GLM and the uniform distribution.

Implementation details for each method can be found in Appendix B. After the GLM is fine-tuned, the class posteriors are computed using Equation (2), as for the unadapted system.

As we will see in the results, the best performance in terms of NER is achieved without further regularization beyond that provided by the LoRA approach, while regularization methods like label smoothing or early stopping result in the best NCE. This poses an apparent dilemma. Which approach should we choose if we care both about having good categorical decisions and also good posteriors? Fortunately, as we will see, we do not need to settle for one or the other, since a combination of SFT and PHC leads to the best performance on both metrics.

### 5.2 Post-hoc calibration (PHC)

PHC approaches consist of adding a final stage to a classification system to transform its outputs, $\mathbf{q}$, into better class posteriors, $\tilde{\mathbf{q}}$ (Silva Filho et al., 2023; Ferrer & Ramos, 2025). In our case, the system's outputs are given by Equation (2), computed with token posteriors obtained from the unadapted or the fine-tuned GLM. One of the simplest calibration methods involves applying an affine transformation to the logarithm of the posteriors of the system, followed by a softmax transform to obtain a new set of posteriors:

$$\tilde{\mathbf{q}} = \text{softmax}(\mathbf{A}\log(\mathbf{q}) + \boldsymbol{\beta}), \tag{6}$$

where $\mathbf{q}$, $\tilde{\mathbf{q}}$, and $\boldsymbol{\beta}$ are vectors of dimension $K$, the number of classes, and $\mathbf{A} \in \mathbb{R}^{K \times K}$. If the parameters $\mathbf{A}$ and $\boldsymbol{\beta}$ of the affine transformation are trained to minimize the cross-entropy, the resulting model will produce the best possible posteriors on the adaptation data. If the transformation does not overfit that data, then those posteriors will also be good on unseen data.

This transform was proposed by Kull et al. (2019) under the name Dirichlet calibration, and under the name matrix, vector and temperature scaling, depending on the form of $A$ and the restrictions on $\beta$, by (Guo et al., 2017). For the binary case, this transform is known as Platt scaling (Platt, 2000). The case where $A$ is a scalar was probably first proposed by (Brummer & Van Leeuwen, 2006) under the name direction-preserving (DP) calibration.[1]

In our experiments, we will use four version of this transformation:

- Vector Scaling (**PHC-VS**): Matrix $\mathbf{A}$ is assumed to be diagonal.

- Direction Preserving (**PHC-DP**): Matrix $\mathbf{A}$ is replaced by a scalar, $\alpha$.

- Bias Only (**PHC-BO**): Same as PHC-DP, but with the weight $\alpha$ fixed at 1.

- Temperature Scaling (**PHC-TS**): Same as PHC-DP but with the bias term fixed at 0.

In all cases, the parameters are trained to minimize the cross-entropy of the training data. We also explore adding a regularization term, as for the SFT methods. In each step, the value of the loss is computed using all calibration samples, and if the value did not decrease for 10 steps, the training is stopped.

In addition to the family of affine PHC methods, we also evaluate the use of Adaptive Temperature Scaling (**PHC-Ada-TS**), proposed by Joy et al. (2023). This approach uses the embedding from the last layer as input to a variational autoencoder (VAE) model that produces the temperature for calibrating each sample.

As we will see in the results, PHC is less effective than SFT in terms of NER for all adaptation sizes. It is also inferior in terms of NEC for the larger adaptation sets. Yet, when the adaptation set is smaller, PHC

---

[1]Some of the methods mentioned take logits instead of log posteriors as input. Yet, it can be easily seen that both expressions are identical by showing that, if $\mathbf{z}$ are the logits, then $\mathbf{q} = \text{softmax}(\mathbf{z})$, and $\text{softmax}(\mathbf{A}\log(\mathbf{q}) + \boldsymbol{\beta}) = \text{softmax}(\mathbf{A}\mathbf{z} + \boldsymbol{\beta})$, since $\log(\mathbf{q}) = \mathbf{z} + c$ and the outer softmax transform normalizes out the shift $c$.

approaches can, in some cases, result in better NCE than SFT. Fortunately, the combined method described below leads to the best performance on both metrics, consistently outperforming or matching the best of both methods on all tested scenarios.

## 5.3 Combining PHC and SFT

SFT can lead to overfitting, even when using LoRA, causing the model to produce suboptimal (overconfident) class posteriors. This overfitting, though, can be solved by applying PHC on the posteriors produced by the model after SFT. However, combining these two approaches is not trivial. The naive approach would be to use the same adaptation samples to fine-tune the model and train the calibrator. This approach does not work since models trained with CE are already well-calibrated on the samples used for training. Hence, training a calibration model using the posteriors produced by the fine-tuned model for the samples used for fine-tuning would lead to a calibration transform close to the identity function. What we wish to fix is the miscalibration that occurs on samples unseen during fine-tuning, i.e., samples with a score distribution that resembles the one we will see during deployment. To this end, ideally, we need two different adaptation sets, one for SFT, and one for PHC. Yet, given a small amount of adaptation data, splitting the data to train the two models is quite suboptimal.

To make the best possible use of the limited amount of adaptation data, we propose the following procedure for combining SFT and PHC:

1. Perform SFT on 70% of the adaptation set, holding out 30% of the samples for the next step, but training until convergence of the training loss.

2. Train the parameters of the calibrator on the samples held out from fine-tuning, using the posteriors produced on that data by the model obtained above. These posteriors are unbiased since those samples were unseen for that model.

3. Perform LoRA-based SFT using all the adaptation samples until convergence. This is the model called "SFT" in Section 5.1.

4. Apply calibration to the predictions of the SFT model from step 3, using the parameters from step 2.

We will refer to this procedure as "SFT + PHC". Note that this method can be implemented with all the PHC variants described in the previous section.

## 6 Experiments

We conducted experiments using the instruction-tuned versions of LLaMA3.2 (Grattafiori et al., 2024), with 1 billion parameters[2], and Qwen2.5 (Yang et al., 2024), with 7 billion parameters[3], on five different classification tasks. The two models were selected to be relatively small to allow for a large number of experiments, while being different in terms of size and performance, with the Qwen model being seven times larger and significantly better than the LlaMA model. The fact that, as we will see, the main conclusions of this work are similar for these two models suggests that the benefit of the proposed approach is not dependent on the particular GLM design choices or its base performance.

In the next sections, we first describe the experimental setup and compare the results using LLaMA3.2 for the various adaptation approaches considered in this work. Then, we explore in detail the impact of the size of the adaptation set on a subset of selected approaches. Finally, we compare the results for the two GLMs.

## 6.1 Experimental set-up

We focus on downstream tasks that can be framed as standard text classification problems, including results for the following five open-access datasets:

---

[2]https://huggingface.co/meta-llama/Llama-3.2-1B-Instruct
[3]https://huggingface.co/Qwen/Qwen2.5-7B-Instruct

- **SST2** (Socher et al., 2013) the Stanford Sentiment Treebank, which includes movie reviews annotated as either positive or negative. We used the GLUE version of the dataset (Wang et al., 2018), which does not contain publicly available labels for the test split, so we used only the training and validation sets. We used 400 samples for evaluation selected from the combination of the original train+validation sets to ensure consistency of the label frequency between training and evaluation sets.

- **AGNews** (Zhang et al., 2015): a collection of news articles grouped into four categories. Samples were downloaded from the `fancyzhx/ag_news` huggingface repository, which contains the original version of the dataset. We used 400 samples for evaluation selected after pooling the original train and test sets.

- **DBPedia** (Lehmann et al., 2015): a dataset derived from Wikipedia, where each article is categorized into one of 14 topics. Samples were downloaded from the `fancyzhx/dbpedia_14` huggingface repository, which contains the standard 14-category version of the dataset. We used 700 samples for evaluation sampled from the original test set.

- **20NewsGroups** (Lang, 1995): posts from online newsgroups categorized into 20 different topics. Samples were downloaded from the `SetFit/20_newsgroups` huggingface repository, which contains a curated variant of the original dataset. We used 800 samples for evaluation selected after pooling the original train and test sets.

- **Banking77** (Casanueva et al., 2020): customer service queries related to online banking, classified into 77 intent categories. Samples were downloaded from the `PolyAI/banking77` huggingface repository, which contains the original version of the dataset. We used 1000 samples for evaluation selected after pooling the original train and test sets.

Appendix A includes details of the prompt template, the class prior distribution, and the number of evaluation samples for each dataset.

We consider the scenario where a small amount of in-domain samples is available for adapting the model to the downstream tasks and study the impact of the adaptation set size in the performance of the different approaches. To this end, we generate adaptation datasets of varying sizes, randomly selecting $N$ samples from the subset not used for evaluation. The value of $N$ was computed as $N = K \cdot 2^T$, i.e., proportional to number of classes, $K$, with the factor being a power of 2. We initially tried using the same set of factors $T$ for all tasks. Yet, we found that the same $T$ resulted in very different trends for tasks with two classes versus tasks with many classes. In particular, while $T = 1$ was too small for SST-2 and AGNews, with 4 and 8 samples being insufficient for adaptation, 154 samples were enough to obtain reasonable results for Banking77. We then concluded that in order to obtain similar trends across datasets we needed to make $T$ depend on the number of classes. Parameterizing $T$ as $T'/\log(K)$ and then rounding the result to the nearest power of 2, resulted in similar trends across datasets for the same value of $T' \in \{4, 5, 6, 7, 8\}$. For the SFT methods that require a validation set, after selecting $N$ adaptation samples, we further split those $N$ samples, keeping 70% for fine-tuning and the rest for validation.

## 6.2 SFT-based adaptation

Figure 2 shows the results for the LLaMA3.2 model after adapting it to each of the datasets above using the SFT methods described in Section 5.1. Each plot shows two groups of bars, corresponding to the training sizes obtained with $T' = 4$ and $T' = 8$. Each bar shows the median NCE or NER across $S$ adapted models obtained by sampling the adaptation data with $S$ different random seeds. We used $S = 5$ seeds for DBPedia, 20Newsgroups and Banking77, for which the results were stable across seeds, and $S = 9$ seeds for SST-2 and AGNews, for which the results were noisier.

The figure shows that the categorical decisions obtained with the unadapted model (NoA) perform better than chance, as indicated by NER values lower than 1.0. On the other hand, the posteriors produced by the unadapted model are extremely poor with NCE values larger than 1.0 for all datasets except SST-2. As expected, SFT provides large gains in terms of both metrics in most datasets even for the smaller adaptation set, though, naturally, the gains are larger for the larger adaptation set. Note that, in these results, SFT is always performed using LoRA, which provides strong regularization compared to full fine-tuning (Biderman et al., 2024). In fact, in our preliminary experiments full fine-tuning produced extremely poor results for

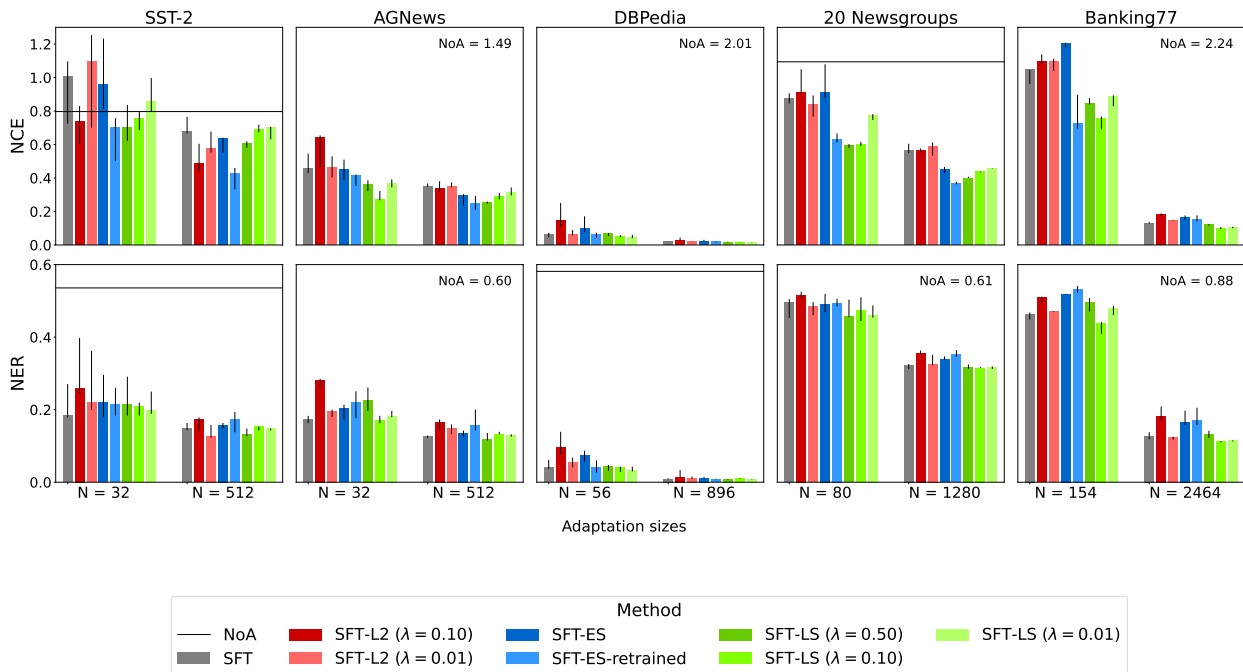

Figure 2: NCE and NER for the unadapted model (NoA) and LoRA-based SFT-adapted models for two adaptation samples (setting $T' = 4$ and $T' = 8$) on LLaMA3.2 model. Bars are divided by groups of colors according to the regularization method used: red for L2 regularization (SFT-L2), blue for early stopping (SFT-ES), green for label smoothing (SFT-LS) and gray for no additional regularization beyond that provided by the LoRA approach. The bar height corresponds to the median across seeds used to select adaptation samples, and confidence intervals (black vertical lines on top of the bars) correspond to the 1st and 3rd quartiles. The NoA values are printed in the top right corner when they fall outside of the y-range.

the smaller amount of adaptation data and no advantages over LoRA-based SFT for the larger adaptation sizes. For this reason, full fine-tuning results are not shown in this paper.

Comparing the baseline LoRA-based SFT results with those that include an additional regularization approach like L2, LS or ES, we can see that the additional regularization provides no benefit in terms of NER, often causing a slight degradation in that metric. In contrast, some regularization approaches – namely, SFT-ES-retrain and SFT-LS – consistently improve the NCE metric. This indicates that LoRA-based SFT still suffers from some degree of overfitting, which affects the quality of the posteriors. The additional regularization provided by the early stopping and the label smoothing methods mitigates this overfitting and, as a consequence, improves the quality of the posteriors which is reflected in a lower NCE.

Comparing SFT-ES with SFT-ES-retrained in terms of NCE, we can see that the retraining step on all the available adaptation data after selecting the number of steps by holding out some validation data (Section 5.1) gives significant gains. For the SFT-LS approach, we see that no single value of $\lambda$ is optimal across all datasets and training data sizes which means that this parameter would have to be tuned for every new use-case scenario. This would require an approach similar to the one used for choosing the number of step for SFT-ES, leaving some validation data to select the $\lambda$ and then retraining on all data using the best value. This would be extremely costly since it would require running SFT-LS several times – one for each value of $\lambda$ to be explored and a final one on the complete adaptation set. Fortunately, as we will see in Section 6.4, the miscalibration produced by SFT can be addressed by applying a post-hoc calibration stage, a more effective and efficient alternative to the regularization approaches compared in this section.

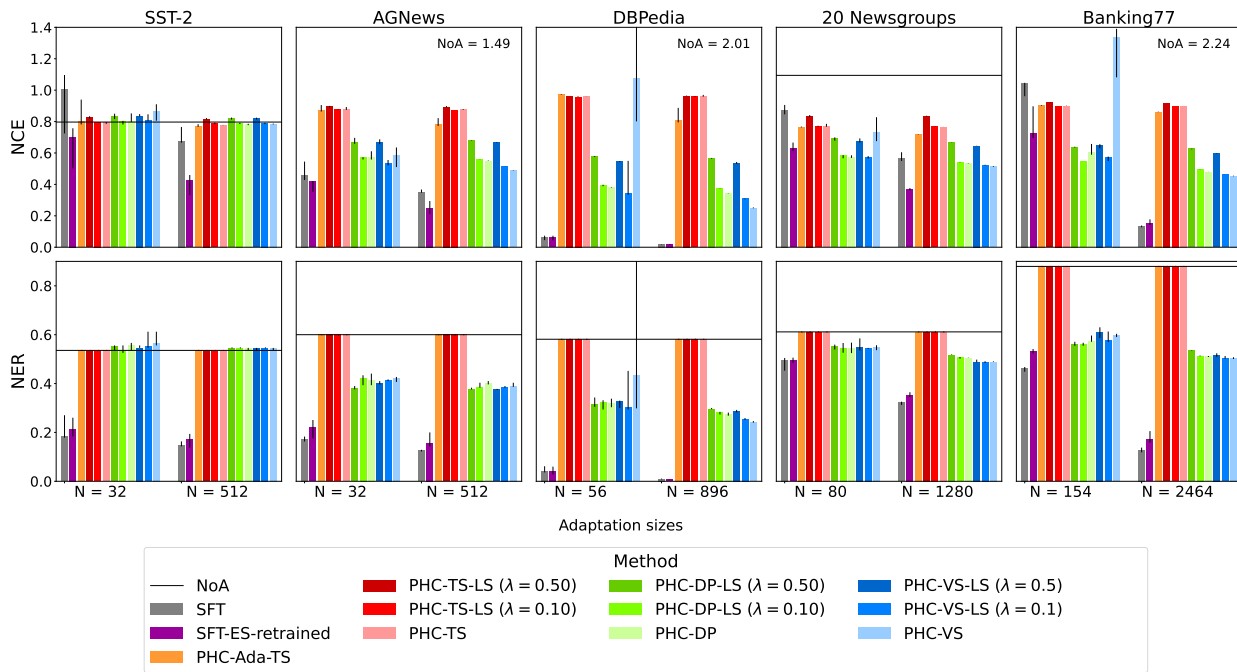

Figure 3: NCE and NER for the unadapted model (NoA) and PHC-adapted models for two adaptation samples (setting $T' = 4$ and $T' = 8$) on LLaMA3.2 model. The type of transform is identified by the color of the bars, while the intensity identifies the degree of label smoothing regularization. We also include results for SFT and SFT-ES-retrained from Figure 2 for reference. The bar height corresponds to the median across seeds used to select adaptation samples, and confidence intervals (black vertical lines on top of the bars) correspond to the 1st and 3rd quartiles. The NoA values are printed in the top right corner when they fall outside of the y-range.

## 6.3 PHC adaptation

Figure 3 shows the results for the PHC methods explained in Section 5.2. The SFT and SFT-ES-retrain results from Figure 2 are also shown for reference. We can see that TS, the most common PHC method in current machine learning literature (Guo et al., 2017), and Ada-TS are significantly worse than DP and VS calibration, both of which include a bias term in addition to a scaling factor. Bias-only (BO) calibration is excluded from the Figure to reduce clutter but results from this method were always in between TS and DP results. As explained by Ferrer & Ramos (2025), the bias term in the calibration transform compensates for mismatches in the prior class distribution between the model and the task of interest, while the scale compensates for overfitting or underfitting problems. The results in Figure 3 suggest that a large source of miscalibration for this GLM is prior mismatch, since the bias term provides significant gains over the scale-only method.

The figure also shows the impact of label smoothing (LS) regularization. This approach was selected because it was superior to the other two regularization approaches we tried: L2 regularization and early stopping. We can see that regularization provides no benefit for TS and DP calibration. On the other hand, for VS, which contains a much larger number of parameters than DP and TS, some degree of regularization is beneficial, particularly for the smaller adaptation sets.

Overall, we can see that PHC-DP and PHC-VS with $\lambda = 0.1$ provide large gains with respect to the unadapted model, but are still generally worse than SFT-adapted models with two exceptions: for the smaller adaptation sizes for 20Newsgroup and Banking77, PHC-DP and PHC-VS provide similar results to SFT.

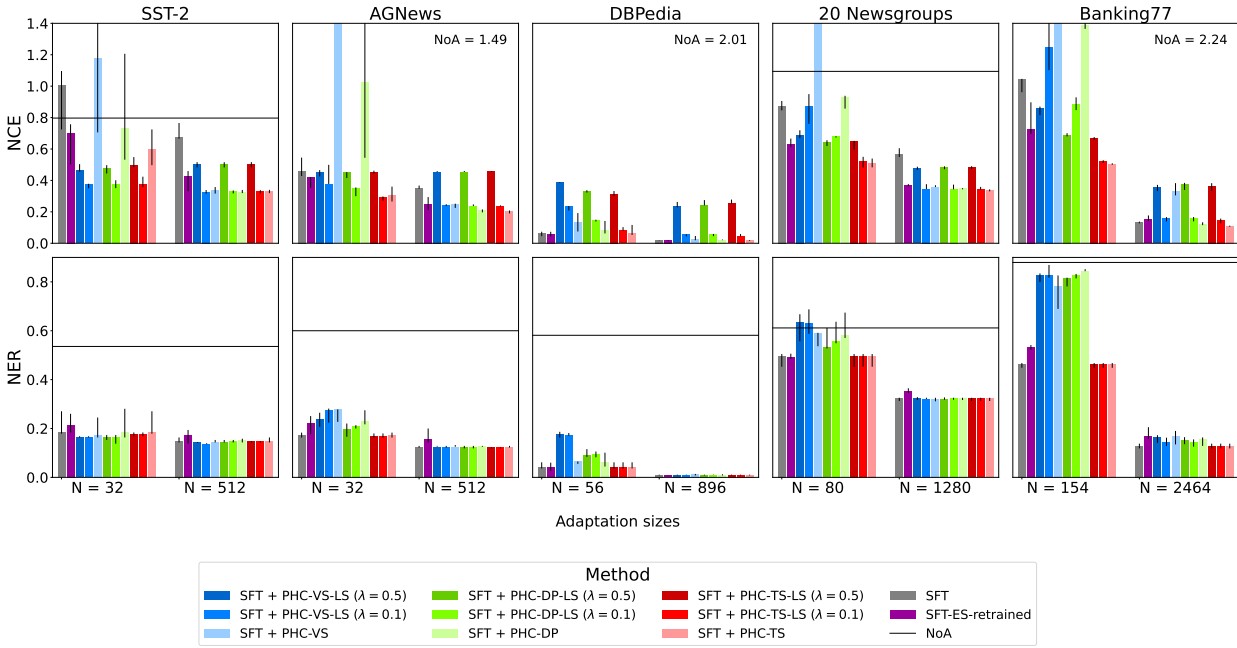

Figure 4: NCE and NER for combined adaptation methods for two adaptation samples (setting $T' = 4$ and $T' = 8$) on LLaMA3.2 model. The color is determined by the type of PHC approach while the intensity depends on the degree of regularization. We include the SFT and SFT-ES-retrain results as reference since they are the overall best performing approaches for NER and NCE, respectively, from Figure 2. The bar height corresponds to the median across seeds used to select adaptation samples, and confidence intervals (black vertical lines on top of the bars) correspond to the 1st and 3rd quartiles. The NoA values are printed in the top right corner when they fall outside of the y-range.

## 6.4 Combined SFT+PHC adaptation

As we saw in Section 6.2, the best categorical decisions as measured by the NER are obtained with LoRA-based SFT without additional regularization. On the other hand, the best posteriors as measured by NCE are obtained by further regularizing the fine-tuning process, indicating that LoRA-based SFT still suffers from some degree of overfitting. While the regularization approaches explored in Section 6.2 improve over the SFT baseline in terms of NCE, they require tuning of the regularization parameter, a very costly endeavor. In this section, we explore the use of PHC as an alternative to regularization to improve the quality of the posteriors obtained with SFT.

Figure 4 shows the performance of adaptation methods that use SFT in combination with PHC using the proposed strategy from Section 5.3, and exploring the use of label smoothing when training the PHC model as in Figure 3. We apply PHC to the model obtained with SFT without additional regularization. We do not show results combining PHC with further regularized versions of SFT because their performance is comparable to or worse than applying PHC to SFT directly. PHC is better at compensating for the overfitting produced during LoRA-based fine-tuning than those regularization approaches, and their combination does not provide any further gain. Finally, adaptive TS is not included in this figure because it gives similar or worse results than standard TS.

Comparing results across the three PHC approaches, we can see that PHC-TS consistently produces the best results for all datasets. This happens in contrast to the results for PHC-only, where a more complex transform like DP or VS was needed. The fact that, after SFT, the best PHC approach is one without a bias term suggests that the mismatch in the priors is fixed by SFT and the only problem left for PHC to fix is the overconfidence in the posteriors produced by overfitting. Importantly, the combined SFT + PHC-TS system

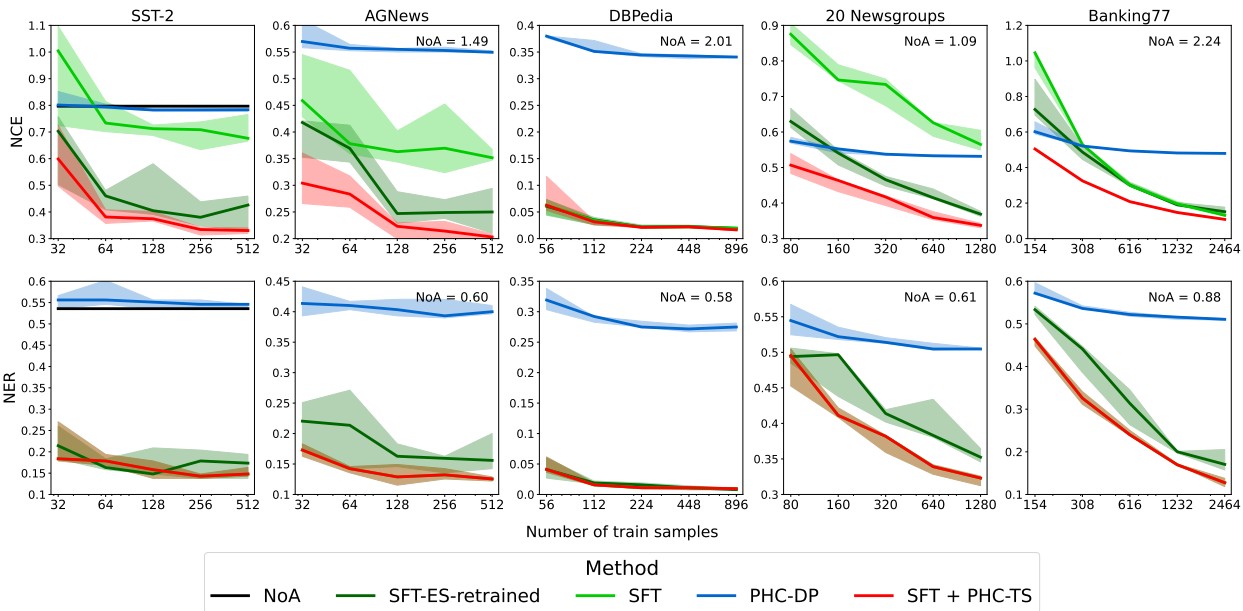

Figure 5: Performance of selected adaptation methods versus the number of adaptation samples on LLaMA3.2. Note that the NER curve for SFT is behind the one for SFT + PHC-TS, since TS does not change the NER value.

is consistently better than or comparable to the best SFT-only approach in terms of NCE (SFT-ES-retrain), while being comparable to the best SFT-only approach in terms of NER (SFT).

Analyzing the impact of regularization, we can see that, for the TS transform, LS helps only in one scenario: SST-2 with the smallest dataset. Notably, regularization helps DP and VS but it does not make them better than TS, strongly supporting the hypothesis that the only aspect to be fixed after SFT is the overconfidence of the posteriors produced by overfitting. This problem can be successfully compensated by a single scale in the log-posterior domain.

In summary, LoRA-based SFT without further regularization followed by PHC-TS consistently gives the best results for both NCE and NER across all the evaluated adaptation approaches, eliminating the need for a trade-off between the two metrics. Further, in addition to performing better, this procedure is also computationally more efficient than applying a further regularization approach during LoRA-based SFT, which requires tuning of the regularization weight or the early stopping epoch. While regularization of the TS transform provides some benefit in one of the test scenarios, properly tuning the $\lambda$ hyperparameter for regularization would require either an additional development set or a computationally-costly cross-validation procedure on the available adaptation data. For this reason and given the marginal gain that regularization of PHC-TS provides, for the rest of the experiments we use the unregularized version. Yet, we note that, when a very limited amount of data is available, it may be worth exploring the use of regularization, using a careful cross-validation approach.

Importantly, the benefit of applying PHC after SFT cannot be obtained if the same data used to train SFT is used to train PHC. This is because, when trained on the same data as SFT, the PHC transform is the identity, since calibration on that data is already perfect. These results are not shown in the plot since they coincide exactly with the SFT results. Hence, the two-stage process proposed in Section 5.3 is essential for achieving a superior model combining SFT and PHC.

## 6.5 Detailed impact of the adaptation set's size

Figure 5 shows the performance of the unadapted model (NoA) and selected adaptation methods for different number of adaptation samples, $N$, obtained by varying $T' \in \{4, 5, 6, 7, 8\}$. As in Figures 3 and 4, we selected

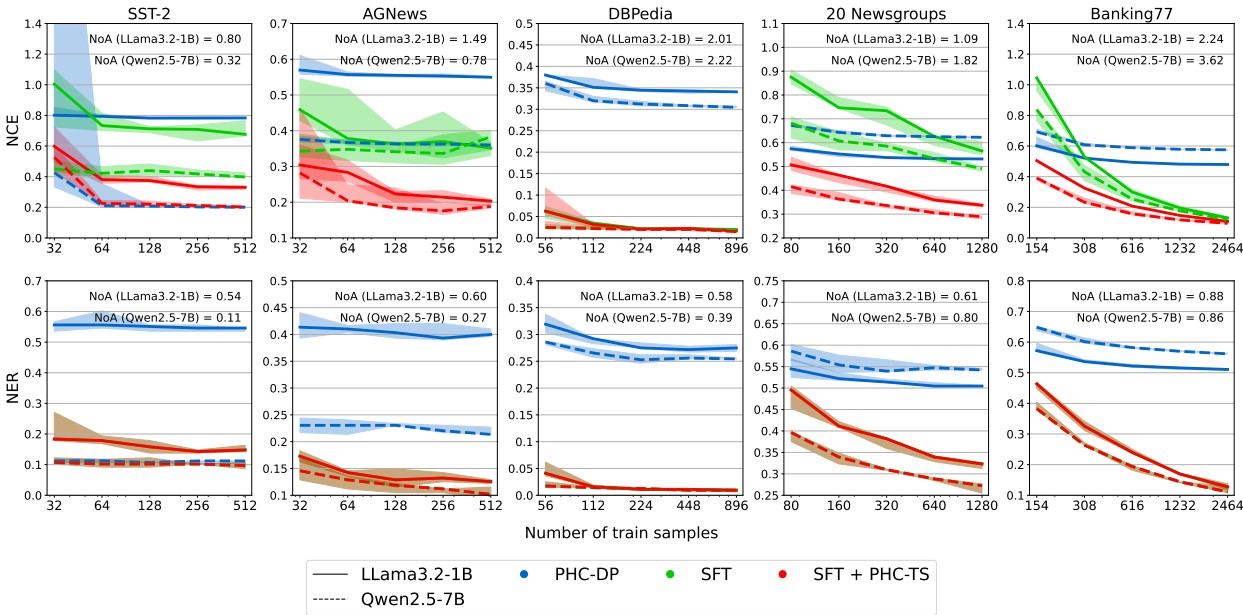

Figure 6: Results for LLaMA3.2 and Qwen2.5 models while varying the number of adaptation samples. Solid lines correspond to the LLaMA model, while dashed lines correspond to the Qwen model.

SFT and SFT-ES-retrain from the fine-tuning methods, for being the overall best methods of their family for NER and NCE, respectively. We also select PHC-DP and SFT + PHC-TS to represent the calibration-only and combined groups, respectively. The results for the left-most and right-most points in each plot coincide with those for the corresponding methods in Figures 2, 3 and 4.

Overall, we can see the expected trend of degradation as the number of adaptation samples decreases for all methods, with a sharper degradation for SFT methods than for PHC-DP. When enough samples are available for adaptation, SFT is significantly better than PHC. However, as the number of samples gets smaller, PHC-DP outperforms SFT-ES-retrain in terms of NCE in two of the five datasets (20Newsgroups and Banking77).

The figure shows that conclusions from the previous section are consistent across all intermediate adaptation sizes: while SFT-ES-retrain is better than SFT for NCE, the trend is reversed for NER. Hence, if we had to select one SFT approach, we would have to prioritize one metric over the other. Further, in some scenarios, PHC-DP gives better NCE than both SFT approaches. Fortunately, the combined approach eliminated the need for a trade-off between metrics, reaching the best results across the board. Table 3 in Appendix C.2 shows the numeric values corresponding to the methods of Figure 5.

## 6.6 Comparison of results for LLaMA and Qwen models

In order to explore whether trends and conclusions hold across different GLMs, we run the main experiments using Qwen2.5-7B-Instruct, a model that is significantly better than the LLaMA3.2 model used in the previous sections. Figure 6 shows a comparison of results for LLaMA3.2 and Qwen2.5 as a function of the number of adaptation samples. The line style identifies the model, while the methods are represented with the same colors as in Figure 5. Here we only include PHC-DP, SFT and SFT + PHC-TS for clarity.

The figure shows that, for the two tasks with fewer classes (SST-2 and AGNews), unadapted Qwen2.5 outperforms unadapted LLaMA3.2, both in terms of quality of decisions and of quality of posteriors. However, for the tasks with more classes (DBPedia, 20Newsgroups, and Banking77) Qwen2.5 slightly underperforms compared to LLaMA3.2, with both models performing considerably worse than the naive baseline in terms of NCE. After SFT or the combined method, Qwen2.5 outperforms LLaMA3.2 consistently, though only by

a relatively small margin. These results suggest that, after adaptation, different models converge to similar performance, despite being initially different.

Importantly, as shown in Figure 10 in Appendix C.3 which presents additional results on the Qwen2.5 model, the main qualitative conclusions are the same as for the LLaMA3.2 model. PHC-DP outperforms PHC-TS, in most cases giving large gains with respect to the unadapted model. SFT-ES-retrain outperforms SFT in terms of NCE, while SFT has a small advantage in terms of NER. Finally, the main result in this paper holds: SFT followed by PHC-TS gives the best results of all adaptation methods across all combinations of dataset, adaptation size, and metric. These results support the hypothesis that the main conclusions in this work are robust to the choice of GLM.

## 7 Conclusions

In this work, we developed and compared various approaches for adapting GLMs to downstream classification tasks. Our main focus was to develop an approach that can produce good-quality class posteriors without sacrificing the quality of the decisions. These posteriors could later be used to make optimal decisions using Bayes decision theory, or simply provided to the user for interpretation. This is particularly important for high-stakes applications, where accepting an incorrect decision may have severe consequences.

We evaluated the performance of the resulting adapted models in terms of Normalized Error Rate (NER), which measures the quality of categorical decisions, and in terms of Normalized Cross Entropy (NCE), which measures the quality of the class posteriors. We compared simple post-hoc calibration approaches that transform the normalized sequence posterior from the GLM into better class posteriors and supervised fine-tuning using LoRA.

Our results show that the normalized sequence posterior from the unadapted model is an extremely poor class posterior, showing worse performance than a naive system that does not have access to the input text and only knows the class priors. Post-hoc calibration greatly improves the quality of the posteriors and also, as a consequence, the quality of the categorical decisions, offering an efficient and effective adaptation approach. While fine-tuning leads to larger improvements than calibration in most cases, it is prone to overfitting, resulting in posteriors that overestimate the model's certainty. To address this issue, we evaluated the use of further regularization approaches in addition to LoRA, showing that label smoothing and early stopping can improve the quality of the posteriors, at the cost of requiring an additional tuning stage for the regularization hyperparameter. Finally, we proposed a combined method where a post-hoc calibration model is applied to the output of the fine-tuned model, relying on an ad-hoc algorithm to optimally leverage the limited amount of adaptation data for training both stages. The model resulting from this combined adaptation approach consistently provides the best performance across all datasets for NCE and NER over a range of adaptation set sizes. In particular, this combined model outperforms the regularized SFT model, while also being computationally lighter, not requiring a tuning stage for a regularization parameter.

The proposed combined approach constitutes a straight-forward and effective approach for adapting a GLM to a specific text classification task using a limited amount of labeled adaptation data. In future work, we plan to explore the integration of prompt engineering techniques for task adaptation, as we believe they could be complementary to the approaches studied in this work and further enhance model performance. In addition, we will expand this study to include more complex tasks like factual question answering, and information retrieval, where a posterior for correctness can be obtained with approaches similar to the ones studied in this work.

**Broader Impact Statement and Limitations**

The methods proposed in this work aim to improve the quality of the class posteriors obtained from generative language model (GLM) for text classification tasks, making their outputs more interpretable and reliable for downstream applications. This has the potential to enhance decision-making in critical areas such as healthcare, finance, and legal applications, where incorrect predictions could have grave consequences. By providing a better measure of uncertainty, our approach could contribute to safer and more reliable AI systems.

It is important to note that the conclusions of this work are limited to cases where adaptation and evaluation scenarios are matched: the same prompt templates are used in both steps and the samples are extracted from the same domain. Under these assumptions, we believe the proposed adaptation approach should work well across different models and domains. If these assumptions do not hold – for example, if the prompt is changed to provide examples of the task during evaluation but not during adaptation – the quality of the posteriors for the adapted system is likely to degrade compared to the matched case. More research is required to determine the degree to which a model adapted using a given prompt or domain can be used for a different prompt or on a different domain.

**Acknowledgments**

Research reported in this work was supported by a JPMorgan Chase Faculty Research Award and an Amazon Research Award Fall 2023. Any opinions, findings, and conclusions or recommendations expressed in this material are those of the author(s) and do not reflect the views of Amazon.

**Disclaimer**

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

# A Dataset details

## A.1 Used prompts

Table 1 shows the structure of the prompt used as input to each model. Table 2 shows the instruction and list of class identifiers for each dataset.

| Model | $\rho(\text{input text}, y)$ |
|---|---|
| LLaMA3.2 | `<|begin_of_text|><|start_header_id|>system<|end_header_id|>` |
| | `instructions and task description<|eot_id|><|start_header_id|>user<|end_header_id|>` |
| | `input text<|start_header_id|>assistant<|end_header_id|>` |
| | `class identifier`$(y)$ |
| Qwen2.5 | `<|im_start|>system` |
| | `instructions and task description<|im_end|>` |
| | `<|im_start|>user` |
| | `input text<|im_end|>` |
| | `<|im_start|>assistant` |
| | `class identifier`$(y)$ |

Table 1: Structure of the prompt for each model

## A.2 Priors distribution

Figure 7 shows the prior distribution for each dataset. Classes were sorted by probability value. SST-2 and AGNews are almost perfectly balanced. For DBPedia and 20Newsgroup, the ratio between the prior for the most frequent and for less frequent class is around 2, while for Banking77 is around 3.

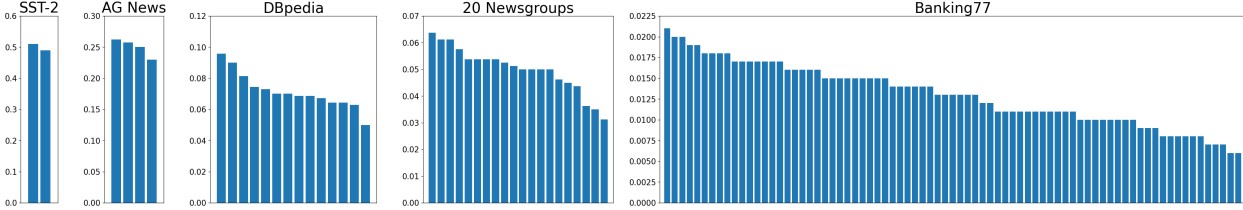

Figure 7: Prior distribution for each dataset. The class names are not included in the x-axis since the main goal is to show the class imbalance in each set.

# B Implementation details

When using SFT in all its variants, we applied LoRA reparametrization to all attention, feed forward layers, and embeddings matrices of the model. LoRA hyperparameters were set to the default ones from the LitGPT library[4] ($p_{\text{dropout}} = 0.05$, $r = 8$, $\alpha = 16$). AdamW optimization was used, with a fixed learning rate of $10^{-3}$ and batch size of 8 for all datasets and adaptation sizes. Training and inference were done in bfloat16 precision.

For SFT-ES, we select the model for which the validation cross-entropy is minimum using early stopping with a patience of 10. For SFT-ES-retrain we proceed as follows: we keep track of the numbers of training steps used for fine-tuning for SFT-ES. Then, we performed a second fine-tuning process with the same parameters and same number of steps, but using all available samples for training. Finally, for SFT we trained the

---

[4] https://github.com/Lightning-AI/litgpt

| Dataset | instruction template | $a_k = \texttt{class-identifier}(y_k)$ |
|---------|---------------------|------------------------------------------|
| SST-2 | Determine if the following review is positive or negative, based on the input given by the user. | *Positive; Negative* |
| AG News | Determine the category of the news article given by the user. | *World; Sports; Business; Science and Technology* |
| DBPedia | Determine the category of the article given by the user. | *Company; Educational Institution; Artist; Athlete; Office Holder; Mean Of Transportation; Building; Natural Place; Village; Animal; Plant; Album; Film; Written Work* |
| 20Newsgroups | Determine the category of the posted document given by the user. | *Atheism; Graphics; Microsoft; IBM Hardware; Mac Hardware; X Window System; Sales; Cars; Motorcycles; Baseball; Hockey; Cryptography; Electronics; Medicine; Space; Christianity; Guns; Middle East; Politics; Religion* |
| Banking77 | Classify the intent of the question input by the user. | *Active my card; Age limit; Apple pay or google pay; ATM support; Automatic top up; Balance not updated after bank transfer; Balance not updated after cheque or cash deposit; Beneficiary not allowed; Cancel transfer; Card about to expire; Card acceptance; Card arrival; Card delivery estimate; Card linking; Card not working; Card payment fee charged; Card payment not recognised; Card payment wrong exchange rate; Card swallowed; Cash withdrawal charge; Cash withdrawal not recognised; Change pin; Compromised card; Contactless not working; Country support; Declined card payment; Declined cash withdrawal; Declined transfer; Direct debit payment not recognised; Disposable card limits; Edit personal details; Exchange charge; Exchange rate; Exchange via app; Extra charge on statement; Failed transfer; Fiat currency support; Get disposable virtual card; Get physical card; Getting spare card; Getting virtual card; Lost or stolen card; Lost or stolen phone; Order physical card; Passcode forgotten; Pending card payment; Pending cash withdrawal; Pending top up; Pending transfer; Pin blocked; Receiving money; Refund not showing up; Request refund; Reverted card payment?; Supported cards and currencies; Terminate account; Top up by bank transfer charge; Top up by card charge; Top up by cash or cheque; Top up failed; Top up limits; Top up reverted; Topping up by card; Transaction charged twice; Transfer fee charged; Transfer into account; Transfer not received by recipient; Transfer timing; Unable to verify identity; Verify my identity; Verify source of funds; Verify top up; Virtual card not working; Visa or mastercard; Why verify identity; Wrong amount of cash received; Wrong exchange rate for cash withdrawal* |

Table 2: Instructions and list of class identifiers for each dataset.

model using all the available samples until the training cross-entropy converged using the same criterion as for SFT-ES.

We train all the PHC methods described in Section 5.2 with the LBFGS algorithm in a single batch per epoch until the training loss does not decrease its value for 10 consecutive steps. Learning rate was set to $10^{-3}$ and the maximum number of iterations in each step of the LBFGS algorithm to 40.

### B.1 Training curves example

Figure 8 shows one seed of the training curves for the SFT-ES method. It can be seen that the validation curve reaches convergence for the chosen patience value (10) for both sizes and all datasets. Note that the model with the best validation loss is selected after training is stopped. Other seeds show the same trend.

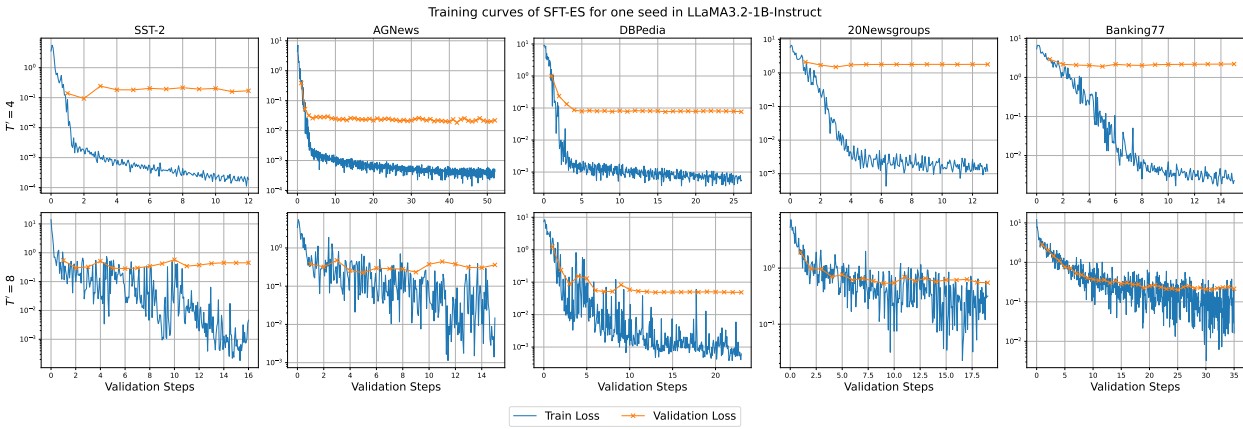

Figure 8: Training curves example for SFT-ES using a patience value of 10.

## C   Additional results

### C.1   Additional metrics

One of the main concerns about the NCE is that the logarithmic loss is an unbounded metric which may allow a few extreme values to disproportionately affect the average. While, as we argue in Section 4, we believe this is a desirable property of this metric, it is interesting to analyze the impact of this behavior on the results. To this end, we evaluated a subset of the methods using the Normalized Brier Score (NBS), which is another PSR that is also commonly used as a metric to assess the quality of posterior probabilities. The NBS, unlike the NCE, is a bounded metric defined as:

$$\text{NBS} = \frac{\sum_{i=1}^{N} \sum_{k=1}^{K} (q_k^{(i)} - \mathbb{1}_{\{y^{(i)}=y_k\}})^2}{\sum_{k=1}^{K} N_k(1 - N_k/N)}, \tag{7}$$

Figure 9 shows the results for a subset of adaptation methods for the LLaMA3.2 model for NBS and for NCE, for comparison. The results for NBS are qualitatively similar to those for NCE. In particular, the main conclusion from the paper – that SFT + PHC-TS is the method that leads to the best results – also holds for the NBS metric. One notable difference in conclusions between the two metrics is that, while SFT is consistently worse than SFT-ES-retrain in terms of NCE, the two methods are comparable in terms of NBS. This is due to the fact that the overfitting that occurs with the SFT method results in overconfident systems that may produce extremely incorrect posterior values, which are much more heavily penalized by the NCE metric. This, in fact, supports our decision to use the NCE metric, since we wish to severely penalize systems that may produce extremely incorrect posteriors.

Figure 9 also shows the Relative Calibration Loss (RCL) (Ferrer & Ramos, 2025) and the ECE (Guo et al., 2017), two metrics designed to measure a model's degree of miscalibration. The RCL is given by the relative difference between the cross-entropy of the model and the cross-entropy that is obtained after applying a calibration transformation. For the results in this figure, the calibrator is given by the DP transform trained on the test set itself. The ECE is given by the absolute difference between the model's posteriors and the posteriors after calibration with histogram binning, also trained on the test data.

The calibration metrics show that the NoA system is greatly miscalibrated, with RCL values between 0.3 and 0.6 (between 30% and 60% of the cross-entropy value is due to miscalibration). The SFT system is also

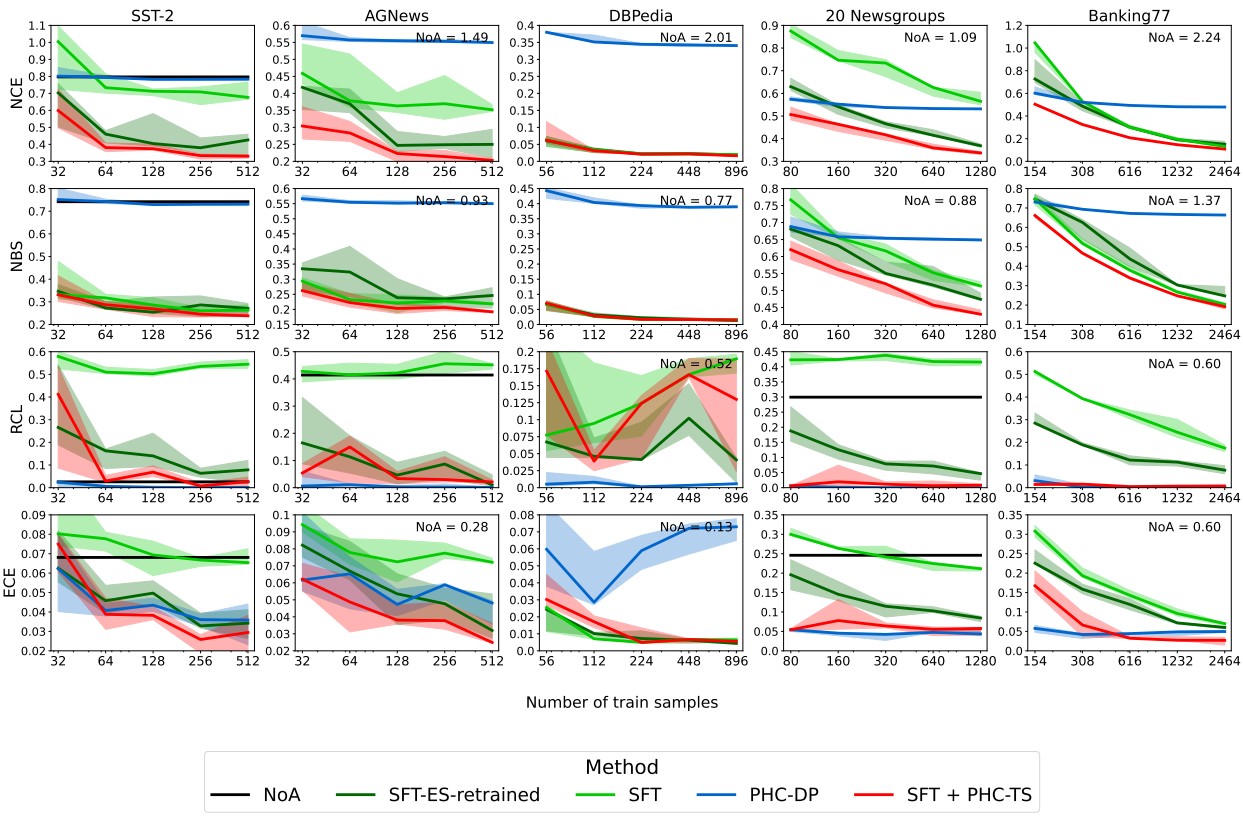

Figure 9: Comparison of NCE, NBS, Relative Calibration Error and ECE metrics for a subset of adaptation methods.

greatly miscalibrated, except for the DBPedia dataset. The other models: SFT-ES-retrained, PHC-DP and the combined model SFT+PHC-TS are all relatively well-calibrated. In particular, the combined system has RCL values below 0.2 for all cases.

The ECE values generally agree with the RCL values, with some notable exceptions. For example, for DBPedia, the NoA system has an RCL of 0.52, indicating gross miscalibration (52% of the cross-entropy is due to miscalibration). Yet, the ECE is 0.13 which may appear relatively low. The discrepancy may be due both to the lack of normalization of the ECE metric and to a suboptimality in the histogram binning transform. An exhaustive discussion on the suboptimality of the ECE can be found in (Ferrer & Ramos, 2025).

In the machine learning literature, it is very common to use calibration metrics instead of SPSRs to assess the quality of posteriors. Figure 9 is an example of why this is a problematic practice. A system may be worse calibrated than another but have a lower cross-entropy. For example, while PHC-DP is close to perfectly calibrated for all datasets, its NCE is worse than the one of the SFT-ES-retrained model. That is, SFT-ES-retrained has better-quality posteriors than PHC-DP despite them being worse calibrated. Hence, we should always use an SPSR to assess the quality of a model's posteriors. Calibration metrics may be used to analyze whether the posteriors can be improved by a post-hoc calibration stage, but the performance after the calibration stage is added should be judged based on an SPSR. An extensive discussion on these issues can be found in (Ferrer & Ramos, 2025).

## C.2 Supplementary table for LLaMA3.2 model

Table 3 shows numerical values of the NCE and NER values for the methods shown in Figure 5. These values correspond to the left-most and right-most points of that figure.

| | | SST-2 | | AGNews | | DBPedia | | 20 Newsgroups | | Banking77 | |
|---|---|---|---|---|---|---|---|---|---|---|---|
| | | NCE | NER | NCE | NER | NCE | NER | NCE | NER | NCE | NER |
| | NoA | 0.80 | 0.54 | 1.49 | 0.60 | 2.01 | 0.58 | 1.09 | 0.61 | 2.24 | 0.88 |
| $T' = 4$ | SFT | 1.00 | **0.18** | 0.46 | **0.17** | **0.06** | **0.04** | 0.88 | 0.50 | 1.05 | **0.46** |
| | SFT-ES-retrained | 0.70 | 0.21 | 0.42 | 0.22 | **0.06** | **0.04** | 0.63 | **0.49** | 0.73 | 0.53 |
| | PHC-DP | 0.80 | 0.56 | 0.57 | 0.41 | 0.38 | 0.32 | 0.57 | 0.54 | 0.60 | 0.57 |
| | SFT + PHC-TS | **0.60** | **0.18** | **0.30** | **0.17** | **0.06** | **0.04** | **0.51** | 0.50 | **0.50** | **0.46** |
| $T' = 8$ | SFT | 0.68 | **0.15** | 0.35 | **0.13** | **0.02** | **0.01** | 0.57 | **0.32** | 0.13 | **0.13** |
| | SFT-ES-retrained | 0.43 | 0.17 | 0.25 | 0.16 | **0.02** | **0.01** | 0.37 | 0.35 | 0.15 | 0.17 |
| | PHC-DP | 0.78 | 0.55 | 0.55 | 0.40 | 0.34 | 0.27 | 0.53 | 0.50 | 0.48 | 0.51 |
| | SFT + PHC-TS | **0.33** | **0.15** | **0.20** | **0.13** | **0.02** | **0.01** | **0.34** | **0.32** | **0.11** | **0.13** |

Table 3: Median value of NCE and NER for the methods shown in Figure 5, corresponding to the LLaMA3.2 model for the smallest ($T' = 4$) and largest ($T' = 8$) adaptation size. The best performance for each task within the corresponding adaptation size is shown in bold.

## C.3 Results for Qwen2.5

Figure 10 shows a selection of the adaptation methods for the Qwen2.5 model. In addition, Figure 11 shows the same results as Figure 5 but for Qwen2.5. As explained in Section 6.6, this model shows better performance than LLaMA3.2 before adaptation and similar performance after adaptation with the proposed method. Note that the NER curves for SST-2 appear noisy but are almost constant (note the range of the y axis). Finally, Table 4 is equivalent to Table 3 but for the Qwen2.5 model.

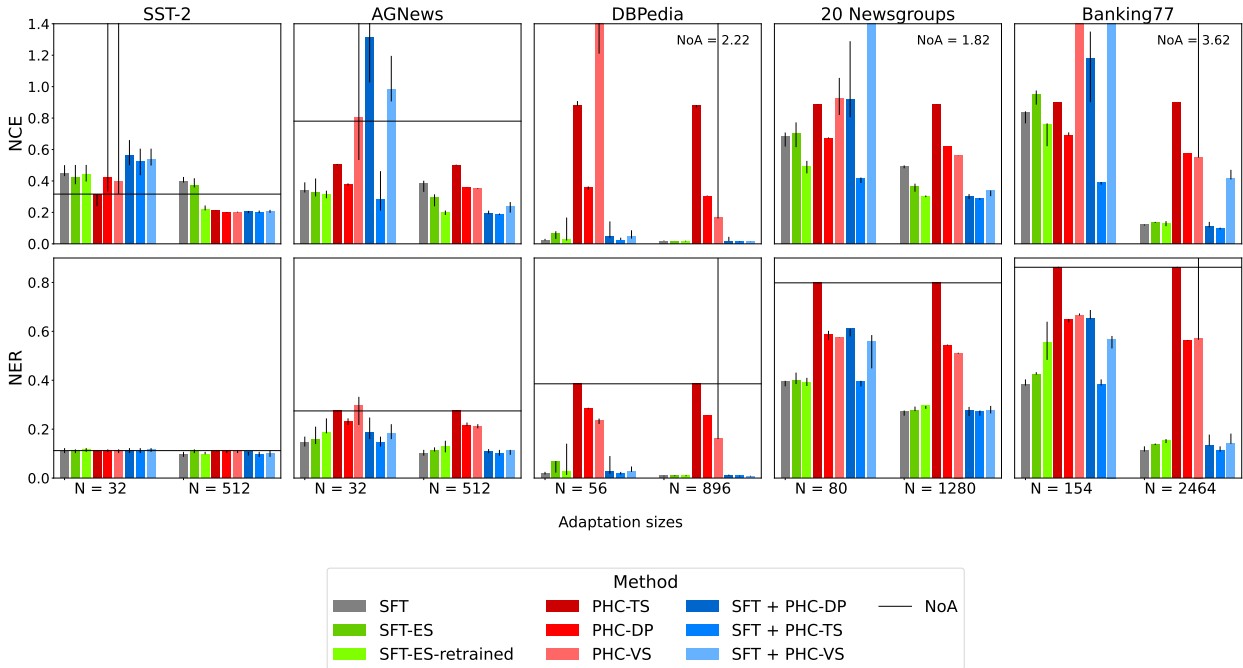

Figure 10: NCE and NER for all adaptation methods and two different adaptation samples ($T' = 4$ and $T' = 8$) on Qwen2.5 model. Bars are divided by groups of colors: green for SFT, red for PHC and blue for combined methods. The bar height corresponds to the median across seeds used to select adaptation samples, and confidence intervals (black vertical lines on top of the bars) correspond to the 1st and 3rd quartiles.

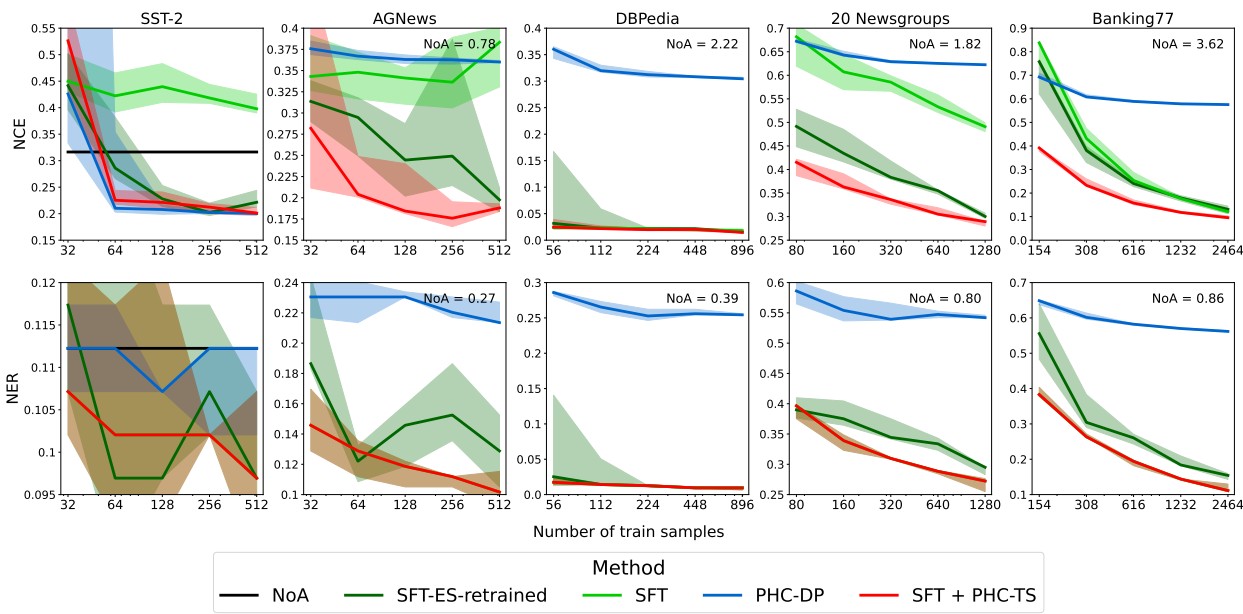

Figure 11: Performance of selected adaptation methods versus the number of adaptation samples on Qwen2.5. Note that the NER curve for SFT is behind the one for SFT + PHC-TS, since TS does not change the NER value. The NER curves for SST-2 appear noisy but are almost constant (note the range of the y axis).

|  |  | SST-2 | | AGNews | | DBPedia | | 20 Newsgroups | | Banking77 | |
|---|---|---|---|---|---|---|---|---|---|---|---|
|  |  | NCE | NER | NCE | NER | NCE | NER | NCE | NER | NCE | NER |
|  | NoA | 0.32 | 0.11 | 0.78 | 0.27 | 2.22 | 0.39 | 1.82 | 0.80 | 3.62 | 0.86 |
| $T' = 4$ | SFT | 0.45 | **0.11** | 0.34 | **0.15** | **0.02** | **0.02** | 0.68 | 0.40 | 0.84 | **0.38** |
|  | SFT-ES-retrained | 0.44 | 0.12 | 0.31 | 0.19 | 0.03 | 0.03 | 0.49 | **0.39** | 0.76 | 0.56 |
|  | PHC-DP | **0.43** | **0.11** | 0.38 | 0.23 | 0.36 | 0.29 | 0.67 | 0.59 | 0.69 | 0.65 |
|  | SFT + PHC-TS | 0.53 | **0.11** | **0.28** | **0.15** | **0.02** | **0.02** | **0.42** | 0.40 | **0.39** | **0.38** |
| $T' = 8$ | SFT | 0.40 | **0.10** | 0.38 | **0.10** | 0.02 | **0.01** | 0.49 | **0.27** | 0.12 | **0.11** |
|  | SFT-ES-retrained | 0.22 | **0.10** | 0.20 | 0.13 | 0.02 | **0.01** | 0.30 | 0.30 | 0.13 | 0.15 |
|  | PHC-DP | **0.20** | 0.11 | 0.36 | 0.21 | 0.30 | 0.25 | 0.62 | 0.54 | 0.58 | 0.56 |
|  | SFT + PHC-TS | **0.20** | **0.10** | **0.19** | **0.10** | **0.01** | **0.01** | 0.29 | **0.27** | **0.10** | **0.11** |

Table 4: Median value of NCE and NER for the methods shown in Figure 11, corresponding to the Qwen2.5 model for the smallest ($T' = 4$) and largest ($T' = 8$) adaptation size. The best performance for each task within the corresponding adaptation size is shown in bold.

