# OpenReview forum: "Adapting Language Models to Produce Good Class Probabilities for Classification Tasks"
_TMLR — Accepted by TMLR_

### Review · Reviewer_KMHy · 2025-08-18

**Summary Of Contributions:**

The main contribution of this paper is proposing a method combining SFT and PHC to improve the quality of posterior probabilities when using GLM to deal with text classification problems. During SFT and PHC process, different datasets are used. The feasibility of this method is validated experimentally.

**Additional Comments:**

n/a

**Audience:**

Yes

**Audience Explanation:**

The proposed method enables a small LLaMA-1B model to achieve posterior probability performance in text classification comparable to that of a significantly larger Qwen-7B.

**Claims And Evidence:**

Yes

**Claims Explanation:**

Mostly clear, though some changes are encouraged.

**Requested Changes:**

1. In the abstract, the first half should also emphasize that this paper focuses on text classification. In its current form, it may give the misleading impression that the work can produce posterior probabilities for answer accuracy in non-text-classification tasks.

2. The main contribution of this paper does not actually include Input–Output prompting. Although some related techniques are used, mentioning it in the related works section seems unnecessary.

3. The authors’ proposed method combines SFT and PHC, and the comparative experiments are designed as SFT-only, PHC-only, and the combination of SFT and PHC. However, both SFT and PHC are fairly standard and commonly used methods. For the posterior probability estimation problem in text classification that the authors aim to address, are there any existing alternative methods that could be included for comparison?

---

> ### Author Response · Authors · 2025-09-12
> **Response to reviewer KMHy**
>
> We thank the reviewer for their valuable comments. Please, find our responses and proposed changes below.
>
> **Clarification on the scope of the paper in the abstract**
>
> The abstract states that “In this work, we focus on the problem of generating reliable class posterior distributions for text classification tasks”. We will add a few example tasks to that sentence to prevent confusion, as follows: “In this work, we focus on the problem of generating reliable class posterior distributions for text classification tasks like sentiment, news category and intent classification.”
>
> **Input-Output prompting in the related work**
>
> Input-output prompting methods are one of the most common approaches for adapting GLMs to a given task. We believe it is important to explain to the readers why we did not consider such methods in our experiments and explain that they can actually be combined with our proposed method. For this reason, we would prefer to keep that explanation in the paper.
>
> **Note on alternatives to SFT and PHC**
>
> There are two big families of methods for doing text classification with GLMs: supervised and unsupervised. In this work, we assume that some amount of labelled data for the task of interest is available for training. Hence, we restrict our comparisons to supervised methods. We do not compare with unsupervised approaches since the literature has shown that supervised approaches outperform unsupervised ones for this task. See, for example, the works by Li et al. (2025), Estienne et al. (2024), and Shen et al (2024).
>
> Within the supervised scenario, most papers that use GLMs for text classification involve some form of prompting or in-context learning. Alternatively, a family of papers proposed the use of SFT, though most such papers start from encoder-only language models like BERT rather than a GLM, as in our work. From these two families of approaches, SFT was shown to be more effective than prompting when enough labelled data is available (e.g, Li et al, 2025; Zhang et al., 2025; Bucher et al., 2024).
>
> SFT has been implemented in different ways, though, as mentioned above, mostly for encoder-only LMs. For example, some works use parameter efficient fine-tuning (PEFT) methods (Houlsby et al., 2019) or add a downstream model (Devlin et al., 2018), potentially training only the new parameters. In the extreme, PHC can be seen as a special case of SFT where the upstream model is frozen and its output is fed to a very small downstream model. Due to the small number of trainable parameters, PHC is robust to overfitting and tends to result in well-calibrated posteriors. This approach was used for text classification with GLMs by Estienne et al. (2024).
>
> Among all the possible SFT variants in the literature, we selected a simple SFT approach based on LoRA to ensure effective use of the available training data, and combined it with affine PHC. In future work, we will explore other SFT approaches. As explained in the paper, we do not explore prompting methods since such approaches have a high computational cost during inference, making them impractical in many applications.
>
> The goal of our paper was to develop an approach that simultaneously optimized the quality of the categorical decisions and of the posterior probabilities. The main conclusions of the paper are that: 1) some degree of overfitting is required to achieve the best discrimination, meaning that an expressive version of SFT is first needed to adapt to the task of interest, and 2) since overfitting degrades calibration, a second stage of re-calibration with a very sparse model is required to obtain also good-quality posteriors. We believe that these conclusions would still hold for other variants of SFT we have not yet explored and for their combination with prompting techniques.
>
> We will include a more thorough justification of our choice of methods for experimentation in the next version of the paper.

---

### Review · Reviewer_e5RS · 2025-08-26

**Summary Of Contributions:**

The paper focuses on improving the reliability of class posterior probabilities generated by large generative language models (GLMs) for text classification tasks, particularly in low-resource scenarios. Key contributions include:

1. **Identifying Limitations of Existing Methods**:
   - The naive approach of computing class posteriors from GLM token probabilities yields poor results.
   - Parameter-efficient supervised fine-tuning (SFT, e.g., LoRA) improves decision quality but leads to suboptimal posteriors due to overfitting.
   - Common post-hoc calibration (PHC) methods like temperature scaling are ineffective for GLMs, while affine transformations (e.g., direction-preserving calibration) perform better.

2. **Proposing a Combined Approach**:
   A three-stage strategy combining SFT and PHC is developed to address overfitting in SFT and enhance posterior quality:
   1. Train SFT on a subset of adaptation data, holding out samples.
   2. Train a PHC calibrator on the held-out samples using posteriors from the SFT model.
   3. Re-train SFT on all data and apply the pre-trained PHC calibrator.
   This method outperforms standalone SFT or PHC in both posterior quality (measured by cross-entropy) and decision quality (measured by error rate).

3. **Evaluation Insights**:
   - Proper scoring rules (e.g., cross-entropy) are more reliable than calibration metrics like ECE for assessing posterior quality.
   - After adaptation, smaller GLMs (e.g., LLaMA3.2-1B) can match the performance of larger models (e.g., Qwen2.5-7B), suggesting the approach makes suboptimal models competitive.

**Audience:**

Yes

**Audience Explanation:**

The paper addresses pressing challenges in LLM reliability, low-resource adaptation, and evaluation—all of which are central to TMLR’s focus. It will engage researchers in areas like natural language processing, calibration of probabilistic models, and efficient fine-tuning, making it relevant to a meaningful segment of the journal’s audience.

**Broader Impact Concerns:**

None.

**Claims And Evidence:**

Yes

**Claims Explanation:**

The paper provides detailed experimental setups, including:
- **Diverse Datasets**: Five text classification datasets (SST2, AGNews, DBPedia, 20Newsgroups, Banking77) with varying class counts and imbalance levels, ensuring generalizability.
- **Clear Methodology**: Precise descriptions of adaptation methods (SFT variants, PHC techniques) and evaluation metrics (Normalized Cross-Entropy [NCE] for posterior quality, Normalized Error Rate [NER] for decision quality), with mathematical formulations and implementation details (e.g., LoRA hyperparameters, training procedures) in appendices.
- **Controlled Variables**: Experiments systematically vary adaptation data size (via \(T'\) parameterization) and compare methods under identical conditions (e.g., same models, seeds, and prompt templates), ensuring results are reproducible.

**Requested Changes:**

Weaknesses:

1. **NCE's sensitivity to extreme values**: Since NCE is unbounded, very high NCE values from a few extreme samples might affect the average result. It would be helpful to check if such cases significantly impact the overall evaluation.

2. **Clarity in Figure 1**: Figure 1 shows a sample prompt but doesn’t explicitly highlight the specific tokens used to calculate the posterior. Adding these tokens could make the posterior computation process easier to follow.

3. **Rationale for cross-entropy**: While cross-entropy is a key metric here, a bit more explanation (beyond citations) on why it’s well-suited for assessing posterior quality in this context would strengthen the core argument.

4. **Model comparison choices**: Comparing a small LLaMA model with a larger Qwen model (different architectures) makes it hard to isolate the effect of the adaptation method. A brief note on this choice would help readers interpret the results.


Changes:

1. A quick check on how extreme NCE values affect the averages—this would confirm the metric’s reliability without altering the main findings.

2. Adding the specific tokens used for posterior calculation to Figure 1 would make the example more intuitive.

3. A short, straightforward explanation of why cross-entropy is ideal here (e.g., how it captures posterior quality for high-stakes tasks) would reinforce the metric’s relevance.

4. A brief note on why Qwen was chosen for comparison (e.g., to show generalizability across models) would help contextualize the model size/architecture differences.

These tweaks would enhance clarity and strengthen the work, while preserving its core contributions.

---

> ### Author Response · Authors · 2025-09-12
> **Response 1/2 to reviewer e5RS**
>
> We thank the reviewer for the valuable comments. Following is a response to the issues highlighted in the weaknesses section along with proposed changes to address those concerns.
>
> **NCE’s sensitivity to extreme values**
>
> We agree that this is a valid concern, since the unbounded nature of NCE may allow a few extreme values to disproportionately affect the average. While, as we argue in Section 4, we believe this is a desirable property of this metric, analyzing the impact of this behavior in the results is a very interesting suggestion.
>
> To address this question, we evaluated the methods using the Normalized Brier Score (NBS), which is another PSR that is also commonly used as a metric to assess the quality of posterior probabilities. The NBS, unlike the NCE, is bounded. The results for NBS are qualitatively similar to those for NCE. In particular, the main conclusion from the paper – that SFT-wo-val+PHC-TS is the method that leads to the best results – also holds for the NBS metric. One notable difference in conclusions between the two metrics is that, while SFT-retrain is consistently better than SFT-wo-val in terms of NCE, the two methods are comparable in terms of NBS. This is due to the fact that the overfitting that occurs with the SFT-wo-val method results in overconfident systems that may produce extremely incorrect posterior values, which are much more heavily penalized by the NCE metric. This, in fact, supports our decision to use the NCE metric, since we wish to severely penalize systems that may produce extremely incorrect posteriors.
>
> We believe these results may be interesting to the reader so we will add them in a new appendix section, along with a discussion on the differences and similarities between the NBS and  NCE results.
>
> **Clarity in Figure 1**
>
> Figure 1 has three different colours specifying different parts of the prompt + answer text that is used to extract the scores from the LLM. The tokens used to extract the posteriors are the ones highlighted in blue, i.e, those identifying the class. We will add this clarification in the caption of the figure.

---

> ### Author Response · Authors · 2025-09-12
> **Response 2/2 to reviewer e5RS**
>
> **Rationale for Cross-Entropy**
>
> The motivation for using Cross-Entropy as metric was briefly explained in Section 4. We will expand the explanation in the first paragraph of that section as follows:
>
> > “In this work, we follow the literature on strict proper scoring rules (SPSRs), which were proposed decades ago for the assessment of the quality of posterior distributions (Winkler & Murphy, 1968; Gneiting & Raftery,2007; Bröcker, 2009). The proper scoring rule (PSR) concept is founded on the assumption that rational decisions should be made by minimizing the expectation of a cost function of interest. The cost function assigns a numeric value to every possible decision given the true class of the sample. Given a posterior distribution over the classes for a given sample, the optimal decision is that which minimizes the expectation of the cost function with respect to that distribution (Duda & Hart, 2001). The cost of this optimal decision, called Bayes decision, defines a PSR (Dawid & Musio, 2014; Brummer, 2010). In other words, a PSR is the cost of the optimal decision for the given posterior distribution. When decisions are categorical, this process results in non-strict PSRs that assess the quality of the posteriors for one particular application defined by the cost function. The most widely used non-strict PSR is the error rate of decisions made with the argmax rule, i.e., by choosing the class with the largest posterior.
>
> > Strict PSRs can be obtained by considering decisions that are probability distributions (Brummer, 2010). Alternatively, strict PSRs can be constructed by integrating over a family of non-strict PSRs with non-zero weights across the full probability simplex (Gneiting & Raftery, 2007; Brümmer, 2010), allowing for an intuitive understanding of the strictness property (Ferrer and Ramos, 2025).
>
>
> > Different SPSRs can be obtained by changing the relative weight of each individual cost function within that integral. One example of an SPSR is the Brier loss, for which the expectation over the data is called Brier score (BS) (Brier, 1950), a commonly used metric in some medical applications (Huang et al., 2020; Van Hoorde et al., 2015). Another SPSR is the negative log-loss (NLL). The cross-entropy (CE) is the expectation of the NLL, a metric widely used as an objective function for training deep neural network models, including GLMs (Wei et al., 2021; Raffel et al., 2020; Chung et al., 2024). As CE and BS are SPSRs, minimizing either of them encourages the output of the models to be reliable posterior probabilities. In this work, we will mostly use CE instead of BS because the former penalizes extremely wrong posteriors more heavily. While Brier loss has a maximum penalty of 1, the NLL can be infinite when a posterior of 0 is assigned to the correct class. We believe this is a desirable characteristic for high-stakes applications where errors may have extreme consequences. For comparison, results for BS are included in the appendix.
>
>
> > SPSRs satisfy an essential property: their expectation with respect to a given reference distribution is minimized when the posterior under evaluation coincides with this reference. Hence, SPSRs encourage systems to closely follow the reference distribution. For further details and formal and conceptual explanations on PSRs, we refer the reader to the work by Dawid & Musio (2014), Brummer (2010), and Ferrer and Ramos (2025).”
>
> **Model comparison choices**
>
> We will add the following text in the revised version in order to clarify this decision:
>
> > “The two models were selected to be relatively small to allow for a large number of experiments, while being different in terms of size and performance, with the Qwen model being seven times larger and significantly better than the LlaMA model. The fact that, as we will see, the main conclusions of this work are similar for these two models suggests that the benefit of the proposed approach is not dependent on the particular GLM design choices or its base performance.”

---

### Review · Reviewer_HgzT · 2025-09-28

**Summary Of Contributions:**

**Summary of contributions.**

The authors investigate combining the post-hoc calibration with the parameter-efficient supervised fine-tuning (PEFT) for generative language models (GLMs) to gain reliable posterior distributions. They claim that the PEFT leads to overfitting for GLMs on adaptation data and propose to apply a calibration method to outputs from fine-tuned models. In addition, they discover that applying temperature scaling calibration, a straightforward and widely known calibration method, would lead to poor quality of the fine-tuned model's posteriors, so they study different adaptation data splitting strategies for PEFT and calibration. Experiment results demonstrate that the well-designed adaptation data splitting strategy could consistently enhance a GLM's posterior distribution.

**Strength.**
1. The idea is straightforward and effective for GLMs on text classifications.
2. This work reveals the impact of adaptation data on the post-hoc calibration method with detailed experiments.

**Weakness & detailed comments.**

About the investigation on the PEFT (LoRA SFT), which would lead to overfitting for GLMs on adaptation data.

1. iiuc, the early stopping step is decided by Eq. (5) for **SFT-w-val** and **SFT-retrain**. Why not use the evaluation metrics Eq. (3) and Eq. (4) to choose the early stopping step?
2. Follow 1. Besides the early stopping to prevent overfitting, I suggest the authors should study other regularization techniques, such as  L2 regularization [1], Entropy regularization (confidence penalties) [2], which directly address the overfitting issues. Some representative techniques suggested in the survey paper are also encouraged to add [3].
3. Follow 1., I suggest revealing the early stopping results to check the choice of the patience number (10).

About Post-hoc calibration (PHC) and temperature scaling (TS).

4. I suggest unifying the form in PHC-X by the matrix or vector forms. For example, PHC-VS $\mathbf{A}$ -> $\mathbf{D}$ for the diagonal matrix; PHC-DP $\alpha \mathbf{I}$; PHC-DP $\mathbf{I}$.
5. "The general expression for the affine transform, with A being a full matrix leads to unstable results and is very prone to overfitting given a relatively small amount of adaptation samples." -> First, why do we need to redesign the form $\mathbf{A}$ from the scaler form in [4]. Second, what is *overfitting* of the softmax transform here?
6. Follow 5., if we also encounter the overfitting issue in PHC, how do we prevent it in scenarios with a small amount of adaptation data?
7. Based on this paper, focusing on Platt scaling for the post-hoc calibration, I suggest adding related works such as Dirichlet calibration [5] and Sample-Dependent Adaptive TS [6] to have a diverse and recent view of the calibration method.

About Combining PHC and SFT

8. The authors present the critical issue of selecting the proper adaptation for the SFT and PHC ('miscalibration that occurs on samples unseen during fine-tuning, i.e., samples with a score distribution that resembles the one we will see during deployment.'). However, how do we ensure that the held-out subset of the adaptation set for PHC is unbiased for the calibration in the deployment? Could you provide the theoretical bounds or the empirical experiences/discussions about this?
9. Is that possible to utilize the K-Fold approach for obtaining a better calibrator? For example, use 5-fold holding-out subsets to obtain 5 calibrators for Step 4.

About presentations.

10. As mentioned in 8., the evidence of producing the reliable class probabilities might not be enough for me. I consider that 'Reliable' should be robust for arbitrary scenarios, such as out-of-distribution tasks or distribution shift (domain shift). Maybe the authors could consider the title like 'Empirical Study of Enhancing the Quality of the Posteriors for Generative Language Models in Classification Tasks'
11. The motivation for selecting post-hoc calibration would be strengthened. For example, if we know the overfitting of the SFT fine-tuned models, why do we choose the post-hoc calibration instead of the regularization methods? Please try to highlight the motivation and rephrase in Section 1, Paragraph 4.
12. While the authors mention that they do not use expected calibration error (ECE) because of the previous work ('ECE and other calibration metrics do not adequately address the problem of assessing the value of posteriors'), I consider that the reveal of the ECE is still valuable for assessing the calibration performance for the novel task such as LLM for classification.

- [1] On calibration of modern neural networks (Guo et al., 2017)
- [2] Regularizing neural networks by penalizing confident output distributions (Pereyra et al., 2017)
- [3] Calibration in deep learning: A survey of the state-of-the-art (Wang, 2023)
- [4] Evaluating Posterior Probabilities: Decision Theory, Proper Scoring Rules, and Calibration (Ferrer & Ramos, 2025)
- [5] Beyond temperature scaling: Obtaining well-calibrated multi-class probabilities with dirichlet calibration (Kull et al., 2019)
- [6] Sample-dependent adaptive temperature scaling for improved calibration (T Joy et al. 2023)

**Additional Comments:**

Dear authors, reviewers, and the action editor, I apologize for my late reply to this work. I will ensure my engagement in the following discussions.
Sincerely, reviewer HgzT

**Audience:**

Yes

**Audience Explanation:**

1. The main contribution of this work is revealing the impact of adaptation data on the post-hoc calibration (PHC) method with detailed experiments and studying the combination of the PHC and supervised fine-tuning (SFT) for the generative language models. I think that it is valuable and interesting to the researcher who wants to apply the generative models to different domains.

**Broader Impact Concerns:**

As mentioned in previous parts about TMLR's audience, I think if this paper were extended to enhance the properties of LLM's output for arbitrary scenarios, it would be impactful to cross-domain applications.

**Claims And Evidence:**

No

**Claims Explanation:**

1. **Claim 1.** They propose to apply a calibration method to outputs from fine-tuned models to mitigate the overfitting brought by PEFT. However, they only test on the Post-Hoc Calibration instead of other calibration methods for the PEFT, which seems overclaim to me.
2. **Claim 2.** "Reliable" for the PHC. As mentioned in Weakness & detailed comments, Point 6, this work misses the investigation to ensure the calibration would be stable.
3. **Claim 3.** "Reliable" for the adaptation set. As mentioned in Weakness & detailed comments, Point 8, this work lacks the investigation into the properties of the held-out subset that impact the final results.

**Requested Changes:**

1. Please at least add the motivation and discussions for selecting Post-Hoc Calibration instead of other calibration methods for the PEFT. Or if you can include and compare other calibration methods would be better.
2. Please reveal more results to ensure the calibration is stable. For example, please clarify what 'overfitting' is for the calibrator. How do we identify/evaluate the issues of our clabrator?
3. Please reveal the properties of the held-out subset of the adaptation set for SFT and PHC. For example, the similarity of distributions between two sets or SFT/PHC performances on each class.
4. I still suggest revealing the ECE of your results. Please reveal ECE or give stronger reasons to convince me not to do that.
5. Please identify the position of this work. You may consider the preliminary study of applying the Post-Hoc Calibration method for PEFT or "reliable" for the proposing methods of LLM for diverse scenarios.

---

> ### Author Response · Authors · 2025-10-03
> **Response 1/2 to reviewer HgzT**
>
> We thank the reviewer for their detailed and insightful comments and suggestions. Following, we include a response to each of the issues raised:
>
> 1. *iiuc, the early stopping step is decided by Eq. (5) for SFT-w-val and SFT-retrain. Why not use the evaluation metrics Eq. (3) and Eq. (4) to choose the early stopping step?*
>
> It is correct that the early stopping step is decided by Eq. (5) for SFT-w-val and SFT-retrain. We considered using the evaluation metrics Eq. (3) and Eq. (4) to choose the early stopping step. Unfortunately, this is computationally much more expensive since in order to calculate these values, the probability for each class has to be computed, which requires forcing all possible completions with each of the classes after the prompt rather than only the one in the given input sample (i.e., the correct class). When the number of classes is large, doing this computation for every validation step is quite time-consuming.
>
> 2. *Follow 1. Besides the early stopping to prevent overfitting, I suggest the authors should study other regularization techniques, such as L2 regularization [1], Entropy regularization (confidence penalties) [2], which directly address the overfitting issues. Some representative techniques suggested in the survey paper are also encouraged to add [3].*
>
> We agree with the reviewer and we will explore a few additional regularization approaches for completeness. Due to the cost of running fine-tuning, we will probably have to limit the experiments to a couple of datasets and training data sizes.
>
> 3. *Follow 1., I suggest revealing the early stopping results to check the choice of the patience number (10).*
>
>  We will provide some examples of training curves in a new appendix.
>
> 4. *I suggest unifying the form in PHC-X by the matrix or vector forms. For example, PHC-VS  $A$ -> $D$ for the diagonal matrix; PHC-DP $\alpha I$; PHC-DP $I$*
>
> We thank the reviewer for the suggestion. Yet, we believe the notation would become awkward, particularly for differentiating the case where there is also a bias term from the case in which there is only a weight term.
>
> 5. *"The general expression for the affine transform, with A being a full matrix leads to unstable results and is very prone to overfitting given a relatively small amount of adaptation samples." -> First, why do we need to redesign the form  from the scaler form in [4]. Second, what is overfitting of the softmax transform here?*
>
> The motivation for generalizing the form of the calibration transform to use a full matrix weight instead of a scalar was to explore whether a more expressive transform could produce better calibrated results. Yet, due to the increased number of parameters that need to be trained when using a full matrix as weight instead of a scalar, the final calibration transform can more easily overfit to the training data. Note that it is not the softmax that overfits, but the parameters of the affine transformation.
>
> 6. *Follow 5., if we also encounter the overfitting issue in PHC, how do we prevent it in scenarios with a small amount of adaptation data?*
>
> PHC in itself does not prevent overfitting. Rather, overfitting is prevented (or mitigated) when learning the calibration transform by restricting this transform to have very few parameters. In the extreme, temperature scaling has a single parameter.  As we show in the paper, even for a very small amount of training samples, that single parameter $alpha$ is unlikely to overfit. On the other hand, even DP calibration, which has a number of parameters equal to the number of classes plus one, can overfit in the data scarcity scenarios tested in our paper.
>
> 7. *Based on this paper, focusing on Platt scaling for the post-hoc calibration, I suggest adding related works such as Dirichlet calibration [5] and Sample-Dependent Adaptive TS [6] to have a diverse and recent view of the calibration method.*
>
> We agree that it is interesting to compare with alternative calibration methods. We will run some additional comparisons for the next draft of the paper.

---

> ### Author Response · Authors · 2025-10-03
> **Response 2/2 to reviewer HgzT**
>
> 8. *The authors present the critical issue of selecting the proper adaptation for the SFT and PHC ('miscalibration that occurs on samples unseen during fine-tuning, i.e., samples with a score distribution that resembles the one we will see during deployment.'). However, how do we ensure that the held-out subset of the adaptation set for PHC is unbiased for the calibration in the deployment? Could you provide the theoretical bounds or the empirical experiences/discussions about this?*
>
> Based on our experience on different tasks, domain shift can have a very large impact in calibration performance, usually much more so than on discrimination performance, and the level of degradation is quite unpredictable. The study of the impact of domain shift is out of the scope of this paper, where we consider a  scenario in which data representative of the deployment data is available for adaptation. Note, though, that, while we do not consider the impact of domain shift, we do report the impact that the randomness in the held-out set would have on the results. This was done by repeating each experiment 5 times using different held-out sets, resulting in the shown confidence intervals.
>
> 9. *Is that possible to utilize the K-Fold approach for obtaining a better calibrator? For example, use 5-fold holding-out subsets to obtain 5 calibrators for Step 4.*
>
> This is indeed possible and quite reasonable. The problem with the K-fold approach, though, is that it would require fine-tuning K different models for each run, increasing the computational cost of the method by a factor of K. In addition, the benefit of that approach would probably be limited since our results suggest that training temperature scaling requires very little data, meaning that the additional data provided by the K-fold approach is not likely to result in significant performance gains.
>
> 10. *As mentioned in 8., the evidence of producing the reliable class probabilities might not be enough for me. I consider that 'Reliable' should be robust for arbitrary scenarios, such as out-of-distribution tasks or distribution shift (domain shift). Maybe the authors could consider the title like 'Empirical Study of Enhancing the Quality of the Posteriors for Generative Language Models in Classification Tasks'*
>
> In this study, we use the term “reliable” to refer to “consistently good quality or trustworthy”, as used by, for example, Bröcker, 2009 or Winker and Murphy, 1968. Specifically, when referring to reliable posteriors, we mean posteriors that can be trusted for decision making with Bayes decision theory. We understand that the term “reliable" could also be understood in some contexts as referring to robustness to domain shifts and we will make sure to clarify this issue in the introduction.
>
> 11. *The motivation for selecting post-hoc calibration would be strengthened. For example, if we know the overfitting of the SFT fine-tuned models, why do we choose the post-hoc calibration instead of the regularization methods? Please try to highlight the motivation and rephrase in Section 1, Paragraph 4.*
>
> Note that we do not choose PHC instead of regularization. We tried two different systems that implement early stopping (SFT-w-val and SFT-retrain)  as a form of regularization and compared it with PHC after fine-tuning without regularization. Our results show that avoiding early stopping leads to improved discrimination performance (lower NER). Yet, the unregularized model suffers from poor calibration. Hence, a combination of an unregularized model followed by PHC turns out to provide the best overall results, both in terms of NER and NCE. We will rephrase the motivation to make this clear. Further, we will include other regularization methods for completeness.
>
> 12. *While the authors mention that they do not use expected calibration error (ECE) because of the previous work ('ECE and other calibration metrics do not adequately address the problem of assessing the value of posteriors'), I consider that the reveal of the ECE is still valuable for assessing the calibration performance for the novel task such as LLM for classification.*
>
> We will add the ECE metric in our results.

---

### Author Response · Authors · 2025-10-03
**Global response to all reviewers**

We thank the reviewers for their valuable and detailed feedback. Following we include an overview of the main changes we plan to perform for the revised version of the paper, which we will attempt to finalize within the next two weeks:
* We will run additional experiments implementing alternative regularization methods to compare with the two versions of early stopping presented for the paper, as suggested by Reviewer HgzT.
* In addition, we will include results with alternative calibration methods, as also suggested by Reviewer HgzT.
* We will clarify the scope of the paper in the abstract and the introduction, explicitly mentioning task examples (to respond to Reviewer KMHy’s concern about the task specification) and providing a rigorous definition of the term “reliable” (in response to Reviewer HgzT’s concern about the use of this term).
* As suggested by reviewer KMHy, we will expand the literature review in Section 2 with a more detailed explanation of the supervised and unsupervised families of methods used for classification tasks, as well as the PEFT variants for small adaptation datasets.
* We will address the question from reviewer e5RS about NCE’s sensitivity to extreme values in two ways. First, we will expand the motivation for using this metric in Section 4. Second, we will add results using the Normalized Brier Score (NBS), including a discussion on the differences and similarities with NCE results.
* We will add the Expected Calibration Error (ECE) results, as requested by Reviewer HgzT.

---

### Author Response · Authors · 2025-11-04
**Summary of changes per section (1/2)**

Dear Reviewers and Editor,

We have uploaded a revised PDF of the manuscript that incorporates the reviewers' comments and suggestions. To facilitate your review, the relevant changes with respect to the previous version have been highlighted in red. Below we list the main changes organized by section.

We remain available for any further clarifications. We thank you for your time and constructive feedback.


## Abstract and Introduction

To respond to Reviewer KMHy’s concern about the task specification, we added task examples in the abstract to clarify the intended scope of the paper. Further, in response to Reviewer HgzT, we included in the introduction a definition of the term “reliable” as it is used in this work, along with references where the terms “reliable posteriors” and “good-quality posteriors” are used.

## Section 2 (Previous works)
At the end of this section, we included some additional references related to generation of posteriors for text classification using LLMs. We further justify our decision to use supervised methods by citing recent evidence that supervised methods are more effective for this task than unsupervised ones. These additions were made to address reviewer KMHy’s question about alternative approaches.

## Section 3 (Extracting posteriors from the GLM)
As requested by Reviewer e5RS, we added a note in Figure 1 to clarify which are the tokens used to compute the model’s scores.

## Section 4 (Theory of Proper Scoring Rules)
To address a comment from reviewer e5RS, we significantly expanded the explanation of the original first paragraph of this section to better motivate the use of Proper Scoring Rules to measure the quality of a posterior, and in particular the use of Cross-Entropy as a metric for evaluating the studied approaches.

---

### Author Response · Authors · 2025-11-04
**Summary of changes per section (2/2)**

## Section 5 (Methods description)
As requested by Reviewer HgzT, we included new regularization methods: L2 regularization (called SFT-L2) and Label Smoothing (called SFT-LS). The explanation of these methods is in the third paragraph of Section 5.1.

To clearly distinguish the original regularization methods based on early stopping to the newly added methods, we renamed the original methods. These changes are not highlighted in the text since it is a simple map given by:

- SFT-wo-val —> SFT
- SFT-w-val —> SFT-ES
- SFT-retrain —> SFT-ES-retrained

As also requested by reviewer HgzT we included results for the adaptive temperature scaling method (which we call PHC-Ada-TS) proposed by [1] and the use of regularization in the PHC methods. We also added a note explaining that the Dirichlet calibration map proposed in [2] is equivalent to the calibration map in Equation (6) of our work.

[1] Sample-dependent adaptive temperature scaling for improved calibration (T Joy et al. 2023)

[2] Beyond temperature scaling: Obtaining well-calibrated multi-class probabilities with dirichlet calibration (Kull et al., 2019)

## Section 6 (Experiments)
- Following the suggestion from Reviewer e5RS, we included a short clarification on the choice of GLMs used for our experiments (first paragraph of section 6)

- In order to discuss the new variants suggested  by Reviewer HgzT, we divided the old Section 6.2 in three different sections and old Figure 2 in three different figures:

    - New Section 6.2 contains the analysis of the SFT-only family of methods, aimed at studying the use of different regularization approaches. New Figure 2 shows the results corresponding to these methods. We conclude that SFT-LS provides competitive performance compared to SFT-ES-retrained, our previously best approach in terms of NCE. However, the value of $\lambda$ strongly impacts the performance of the SFT-LS method. The requirement to tune this parameter would increase the computational cost of this method compared to SFT-ES-retrained.

    - New Section 6.3 contains the analysis of the PHC family of methods, which was previously included in the old Section 6.2. Consistently, new Figure 3 includes only the results of the PHC methods, which were previously shown in the old Figure 2. The new Figure 3 now also includes results for different degrees of regularization for each PHC method, which we had not explored before, and for the adaptive TS method.

    - New Section 6.4 contains the analysis of the combination of the SFT and PHC methods using the proposed approach. Results are shown in the new Figure 4 for different LS regularization degrees for the PHC model. We show that SFT trained without further regularization beyond that provided by the LoRA method followed by PHC-TS, consistently gives the best results for both NCE and NER across all the evaluated adaptation approaches. We also note that properly tuning the $\lambda$ hyperparameter for regularization may lead to marginal improvements in extremely low-data scenarios, but at the expense of higher computational requirements due to the need for parameter tuning.

## Appendices:
- We included a new Appendix B.1 with example training loss curves when using Early Stopping, as requested by Reviewer HgzT. All training curves show that the model converges on the validation set when using a patience value of 10.

- In response to Reviewers e5RS and HgzT, we created a new Appendix C.1 to show complementary metrics (Figure 9). First, we show that the main conclusion of the paper – that SFT + PHC-TS provides the best-quality posteriors – still holds when the quality of the posteriors is measured in terms of NBS instead of NCE. Further, we show that NBS fails to diagnose the poor calibration of the SFT model, due to its bounded nature, supporting our decision to use NCE as a metric to assess the quality of posteriors. Finally, we include two metrics that assess the  model’s degree of miscalibration, the ECE and the Relative Calibration Loss (RCL). We show that the combined SFT + PHC-TS has a low calibration error in most scenarios.

---

### Decision · Action_Editor_4x9X · 2025-12-31

**Recommendation:** Accept as is

**Audience:**

Yes

**Audience Explanation:**

All reviewers agree that the findings should be of interest to the TMLR audience.

**Claims And Evidence:**

Yes

**Claims Explanation:**

This paper tackles an important problem: producing reliable class posterior probabilities from language models for text classification, particularly in low-resource settings. The authors convincingly show that naïve posterior extraction is unreliable and that parameter-efficient SFT, while improving accuracy, leads to overconfident and poorly calibrated posteriors.

The main contribution, a three-stage strategy combining SFT with post-hoc calibration (PHC) using a careful data-splitting scheme, is simple, well-motivated, and effective. Experimental results across multiple datasets and two GLMs consistently demonstrate improvements in both decision quality and posterior quality. The use of proper scoring rules is appropriate, and the expanded discussion comparing cross-entropy and Brier score in the revision strengthens the methodological grounding.

The authors have addressed reviewer concerns thoroughly, adding clarifications on metrics, figures, abstract scope, and model comparisons. While the individual components (SFT and PHC) are standard, the paper's value lies in its clear empirical insights into their interaction and its practical guidance for obtaining trustworthy probabilities from GLMs.

Overall, after revision, the claims are well supported with improved clarity and rigor, and the findings should be of interest to the TMLR audience. I recommend acceptance.